# Soil gas radon and soil permeability assessment: Mapping radon risk areas in Perak State, Malaysia

Habila Nuhu[1,2]*, Suhairul Hashim[1,3]☯, Muneer Aziz Saleh[4]☯, Mohamad Syazwan Mohd Sanusi[1‡], Ahmad Hussein Alomari[5‡], Mohamad Hidayat Jamal[6‡], Rini Asnida Abdullah[6‡], Sitti Asmah Hassan[6‡]

1 Department of Physics, Faculty of Science, Universiti Teknologi Malaysia, Skudai, Johor, Malaysia, 2 Department of Science, Plateau State Polytechnic Barkin Ladi Jos, Plateau State, Nigeria, 3 Ibnu Sina Institute for Scientific and Industrial Research (ISISIR), Universiti Teknologi Malaysia, Skudai, Johor, Malaysia, 4 Nuclear Engineering Programme, Universiti Teknologi Malaysia, Skudai, Johor, Malaysia, 5 Energy and Minerals Regulatory Commission (EMRC), Amman, Jordan, 6 Faculty of Engineering, Universiti Teknologi Malaysia, Skudai, Johor, Malaysia

☯ These authors contributed equally to this work.
‡ MSMS, AHA, MHJ, RAA and SAH also contributed equally to this work.
* nuhuhabila@gmail.com

**Data Availability Statement:** All relevant data are within the manuscript and its Supporting Information files.

## Abstract

In this study geogenic radon potential (GRP) mapping was carried out on the bases of field radon in soil gas concentration and soil gas permeability measurements by considering the corresponding geological formations. The spatial pattern of soil gas radon concentration, soil permeability, and GRP and the relationship between geological formations and these parameters was studied by performing detailed spatial analysis. The radon activity concentration in soil gas ranged from 0.11 to 434.5 kBq m$^{-3}$ with a mean of 18.96 kBq m$^{-3}$, and a standard deviation was 55.38 kBq m$^{-3}$. The soil gas permeability ranged from $5.2 \times 10^{-14}$ to $5.2 \times 10^{-12}$ m$^2$, with a mean of $5.65 \times 10^{-13}$ m$^2$. The GRP values were computed from the $^{222}$Rn activity concentration and soil gas permeability data. The range of GRP values was from 0.04 to 154.08. Locations on igneous granite rock geology were characterized by higher soil radon gas activity and higher GRP, making them radon-prone areas according to international standards. The other study locations fall between the low to medium risk, except for areas with high soil permeability, which are not internationally classified as radon prone. A GRP map was created displaying radon-prone areas for the study location using Kriging/Cokriging, based on in situ and predicted measured values. The GRP map assists in human health risk assessment and risk reduction since it indicates the potential of the source of radon and can serve as a vital tool for radon combat planning.

## Introduction

Radon exists in the natural environment and is a matter of concern due to its human health effects. Radon is a radioactive gas that is released from the decay chains of $^{238}$U, $^{232}$Th, and $^{235}$U which are naturally occurring radionuclides. Radon ($^{222}$Rn) is the most abundant isotope of radon and it is the

**Funding:** We acknowledged the financial support from the Ministry of Higher Education Malaysia and Universiti Teknologi Malaysia through UTMSHINE Signature Grant (No. 07G82, 07G85, 07G90 and 07G91). This study was also partially funded by a contract research DTD grant from Intech Scientific Sdn. Bhd. (R.J130000.7617.4C403)

**Competing interests:** The authors have declared that no competing interests exist.

decay product of radium ($^{226}$Ra) in the $^{238}$U decay chain. $^{222}$Rn decays to stable $^{206}$Pb, with some daughters ($^{218}$Po, $^{214}$Po, $^{214}$Pb $^{214}$Bi) in the decay chain which are solids, and reactive. $^{222}$Rn accounts for about 50% of natural radiation dose due to ionizing radiation [1]. Recent studies have attributed cancer incidence to relatively low $^{222}$Rn levels and prolonged exposure [2–5]. The International Agency for Research on Cancer (IARC) rates $^{222}$Rn as a first-class human carcinogen because of it being one of the leading causes of lung cancer after Tobacco [6]. The analysis of $^{222}$Rn in any source (soil, water, building materials) is necessary for the protection of the population from exposure. It is important for such action plans to accommodate the identification of the radon-prone areas (RPA) [i.e areas with the potential to generate higher indoor radon levels] [7].

Generally, two methods are used to designate RPA: (i) direct measurement of indoor radon concentrations and (ii) indirect geostatistical-based methods using proxy indicators (soil gas radon concentration, gamma dose rate from the field, and Aerial measurements of uranium/ radium content in the soils and rocks) [8–11]. In different parts of the world, studies were conducted by determining the soil gas radon concentration to estimate radon-prone areas and radon risk assessment based on geological formations [3, 11–18]. These studies revealed that there is a strong link between radon concentration in soil gas and the underlying geology.

Geogenic radon potential (GRP) is the parameter that quantifies the amount of radon transported from nearby geological formation to the atmosphere or an indicator of the potential of the soil to constitute a source of indoor radon [15]. Several studies have reported a relationship between GRP and indoor radon concentration, while others have obtained radon risk maps based on the measurement of radon activity concentration and soil gas permeability [15, 19–23]. High GRP indicates a high probability of radon entering indoors due to geogenic reasons, such as radium and uranium content of the bedrock and rock permeability as well as faults that are related to radon mobility [14]. There are several methods used to map the GRP of a place. The most used methods are based on measured parameters such as uranium concentration in rocks, radium, and radon in soils, gamma-ray measurements by airborne gamma spectroscopy, and permeability of rocks or soils [17].

Soil gas permeability is a fundamental parameter for the determination of radon gas mobility [24]. The presence of a superficial soil layer with low permeability could imply an increase in the accumulation of radon below and may also lead to a considerable decrease in the radon's soil-atmosphere flux [25]. The radon concentration in soil gas is directly dependent on the geological characteristics of the area and can be strongly influenced by soil permeability [16].

A few localized studies of radon in soil gas concentration have been conducted in different parts of Malaysia [26–33], their results ranged from 0.13 to 375.42 kBq m$^{-3}$. According to the existing literature, the data of Malaysian studies on radon concentration in soil has not yet been fully determined despite its apparent health risk [34]. Limited data are available in the literature for $^{222}$Rn concentration in Perak state Malaysia.

The results of field measurements of radon activity concentration and soil permeability from Perak State, Malaysia, are presented in this paper. The goal of the research is to obtain the GRP map of the study area, which is a useful tool for the estimation of radon risk (highlighting regions of elevated $^{222}$Rn levels) and land use planning purposes. "Radon risk" here refers to the probability or likelihood of harm or the severity of the harm from radon exposure. The harm is mainly lung cancer, which is a painful and fatal disease [35].

## Materials and method

### Study area and sampling points

The study area is located in northern Peninsular Malaysia, and its coordinates are between latitudes 3.50° to 5.35° North, and longitudes 100.00° to 101.75° East, within an area of 21,006

km$^2$ [36, 37]. There are two tectonic blocks in Peninsular Malaysia, the Sibumasu block (western region) and Sukhotai Arc (Eastern Malaya block) that are separated by the Bentong-Raub suture zone. These blocks converge at the late Triassic and form a part of the Sundaland continental core of Southeast Asia [38]. The Sibamasu block is in the North-south extension belt known as the western belt of Peninsular Malaysia also referred to as "The main granite range" [39]. The study area is located on this main granite range of Peninsular Malaysia. It is divided into four major geological regions, that is Quaternary (Comprise of marine and continental deposits, such as clay, silt, sand, peat with minor gravel and basalt of early Pleistocene age) covering 24% of the study area; Triassic-Jurassic (Sedimentary rocks of marine origin and form a wide belt. They entail shale, mudstone, siltstone, sandstone, and minor limestone lenses. Interbeds of tuff are common within this openly crumpled sequence) which covers 15% of the study area; Silurian (consists of tightly compressed consolidated shale, slate, argillite, metasandstone, phyllite, and schist) covering 16% of the study area, and Intrusive rocks (Mainly undifferentiated igneous rocks of granitic origin) which cover 45% of the study area [40, 41]. Fig 1 is the Geological map of Perak state with sampling points. The selection of sampling points was done randomly, based on the geological formation and the soil type typical of the region surrounding the sampling site. Each sampling point was picked on the geological and soil map respectively, and its geographical coordinates were noted and used to check the

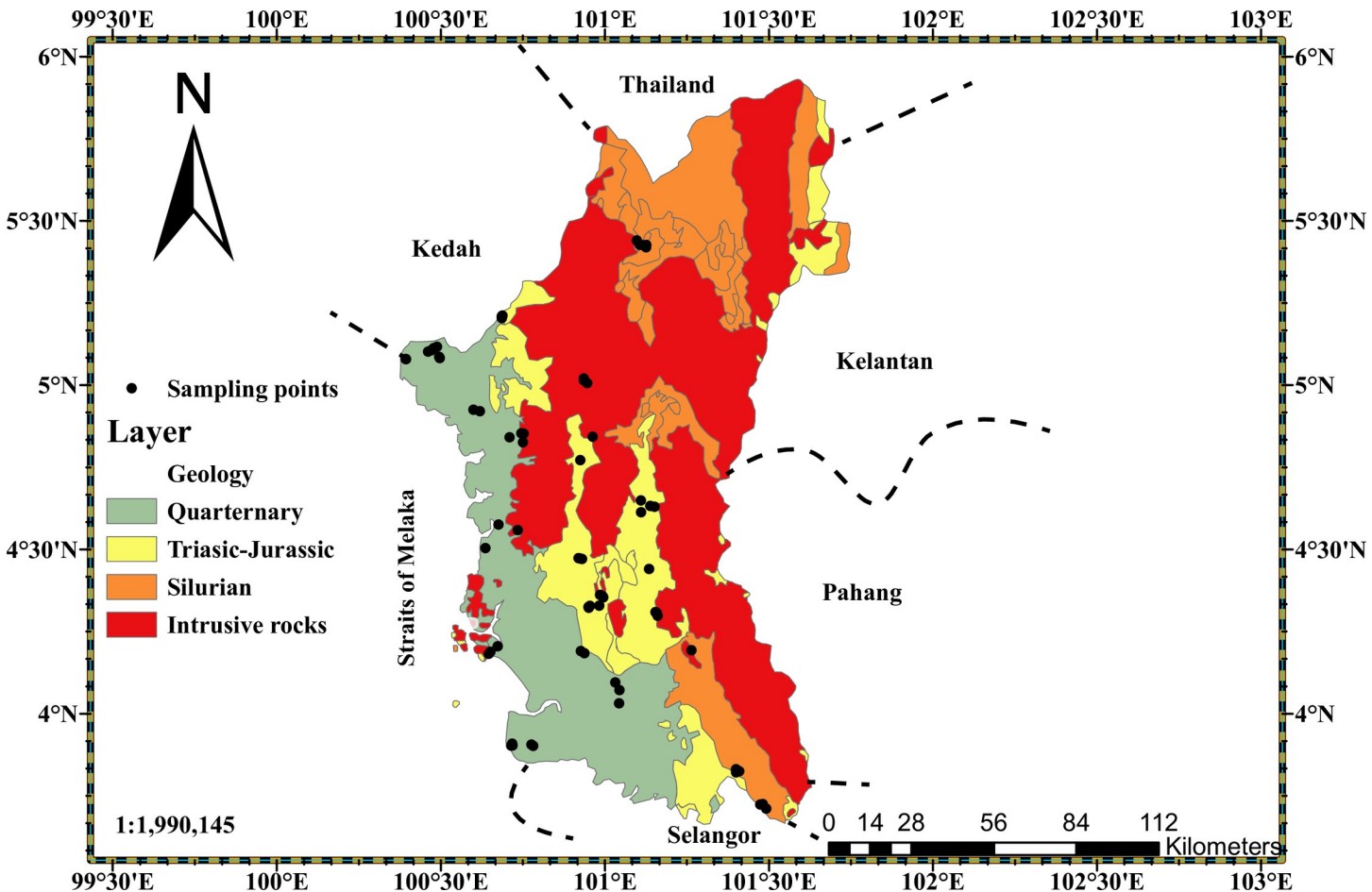

**Fig 1. Geological map of Perak state (digitized from geological map of Peninsular Malaysia 1985 using Arc GIS Pro software).**

accessibility of the site on google map. The selection of sampling location is also free from exclusive or private and personal areas, which required no permit for an on-ground survey. The data of the sampling points were later imported on the map of the geological formation, using the ArcGIS Pro software. The maps were checked to ensure that each geological formation as well as areas with the tendency of radiological magnitudes of radium content and gamma radiation exposure as reported by previous researchers were captured [42]. Thus, seventy (70) locations were sampled to obtain a statistical distribution of Geogenic radon potential representative of the study area.

## Measurement of radon in soil gas

The RAD7 detector was used for the measurement of radon in soil gas. It is an alpha detector, consisting of a solid-state ion-implanted silicon semiconductor, which determines the $^{222}$Rn concentration by measuring the radioactivity of the decay products ($^{218}$Po, and $^{214}$Po), which do reach equilibrium with its parent nuclide in fifteen minutes. The instrument has a low intrinsic background of 0.2 Bq m$^{-3}$, and a measurement range of 4 Bq m$^{-3}$ to 750,000 Bq m$^{-3}$. Measurements were made at a depth of 0.8m (which ensures that there is no dilution with the air from the surface) using a cylindrical probe. The Probe was connected to the RAD7 detector and the test was started using selected commands (GRAB Protocol, SNIFF protocol, or THORON protocol) for the desired measurements [43]. Radon activity concentration measurements were carried out at low or no rainfall periods. The drying unit was used to maintain the relative humidity of the RAD7 at low values.

## Soil gas permeability measurement

The RADON JOK Permeameter was used to measure the soil permeability (k) at each sampling point where radon in soil gas has been measured. The RADON-JOK equipment is used for in-situ measurements of gas permeability of soils. Air is pumped from the soil under constant pressure to a specially designed rubber sack, via gravity with the help of one or two weights. The permeability was calculated based on the time taken for the rubber sack to be filled with air and the use of a nomograph supplied by the Radon V.o.s company. The upper detection limit of the RADON JOK permeameter is 6 s, corresponding to the permeability of $1.8 \times 10^{-11}$ m$^2$. It is recommended that the auxiliary limit for low permeability be set at $5.2 \times 10^{-14}$ m$^2$ when the measuring time is higher than 1200s [23, 44]. In this study, the above detection limits were adapted.

## Ordinary Kriging/Cokriging

Ordinary Kriging/CoKriging is a geostatistical technique used for GIS data interpolation (mapping, zoning, and contouring) purposes. They are both generalized forms of univariate and multivariate linear regression models, for estimation at a point, over an area, or a volume. They are linear-weighted averaging methods, similar to other statistical interpolation methods; however, their weights depend not only on distance but also on the direction and orientation of the neighbouring data to the unsampled location [45].

Ordinary Kriging utilized semivariogram analysis which is a statistical data extrapolation one can use to express autocorrelation between the independent variables either negatively or positively correlated [45, 46]. It also can be used for data transformations and to remove trends due to extreme values, and allow for measurement of residuals to assess the model or map precisions. The estimates are weighting factors calculated from the Kriging data interpolation. The weight factors in kriging are determined by using a user-specified semivariogram model (based on the output of the spatial correlation operation), the distribution of input points, and

are calculated in such a way that minimizes the estimation of error in each output pixel. The estimated or predicted values are thus a linear combination of the input values and have a minimum estimation error. This gives Kriging an advantage over moving averages like the inverse distance [45]. The general probabilistic model of semi-variogram is given as;

$$\gamma(h) = \frac{1}{2n(h)} \sum_{i=1}^{n} \{Z(x_i) - Z(x_i + h)\}^2 \qquad (1)$$

where,

n(h) = Set of pairs of observations i for lag distance h.

$Z(x_i)$ = sampled variable with GIS information (geographic information system).

CoKriging requires GIS data for each data input (e.g., for this work radon gas and soil permeability data) and projects a raster map with estimations of the model residuals, a data interpolation error based on variances of the predicted values from the best line fit [45, 46].

The semivariogram model used in this study is the stable model, which is relatively fast and flexible and balances performance and accuracy. The fitted model line should be close to the averaged crosses on the semivariogram graph. The search neighbour type is standard with a maximum of 15 and a minimum of 10. The geostatistical wizard tool in ArcGIS Pro software was used for this analysis. The resolution of the maps was 300 DPI, and the dimension of the cell size was 5 km × 5 km.

## Statistical analysis

The Origin pro was used to illustrate the data distribution, measures of central tendency, and variability of the measured parameters (radon in soil gas, soil permeability, and geogenic radon potential) in histogram and box plots. The box plot graphically depicts numerical data through their quantiles. The line inside each box represents the median (Q2 or second quartile), while the lower and upper edges of the box are the Q1 and Q3 (first and third quartile) respectively. The vertices of the whiskers stand for the least datum at the range of 1.5 IQR (Interquartile range: Q3 –Q1) of the lower quartile, and the maximum datum at a range of 1.5 IQR of the upper quartile, hence are not considered outliers. Plotting of outliers is done individually at such intervals as dots [47].

# Results and discussion

## Activity concentration of radon in soil gas

Fig 2 displays the histogram with statistical indicators of the radon activity concentration in the study area. The data distribution is asymmetric with lower values having higher frequencies, ranging from 0.11 to 434.5 kBq m$^{-3}$. From the distribution, the mean value was 18.96 kBq m$^{-3}$, while the standard deviation was 55.38 kBq m$^{-3}$. (81%) of the data are below 20 kBq m$^{-3}$, with a few values (13%) of the data having high activity concentration (above 20 to 50 kBq m$^{-3}$), and 6% had very high values (above 50 kBq m$^{-3}$). The Skewness and kurtosis values show that the data is asymmetric and non-normal distribution, prompting log normal distribution which was tested with the Shapiro-Wilk test (P < 0.05). The data were log-transformed to avoid the effect of remarkably high values on the geostatistical analysis. The coefficient of determination (r$^2$) of the log-normal distribution was 0.89. The result did not reject the null hypothesis at a 5% level of significance, that the data distribution complies with log-normal distribution, having a p-value of 0.38. This theoretical distribution is generally used in geostatistical analysis, in the study of the spatial distribution of minerals and soil formations [48].

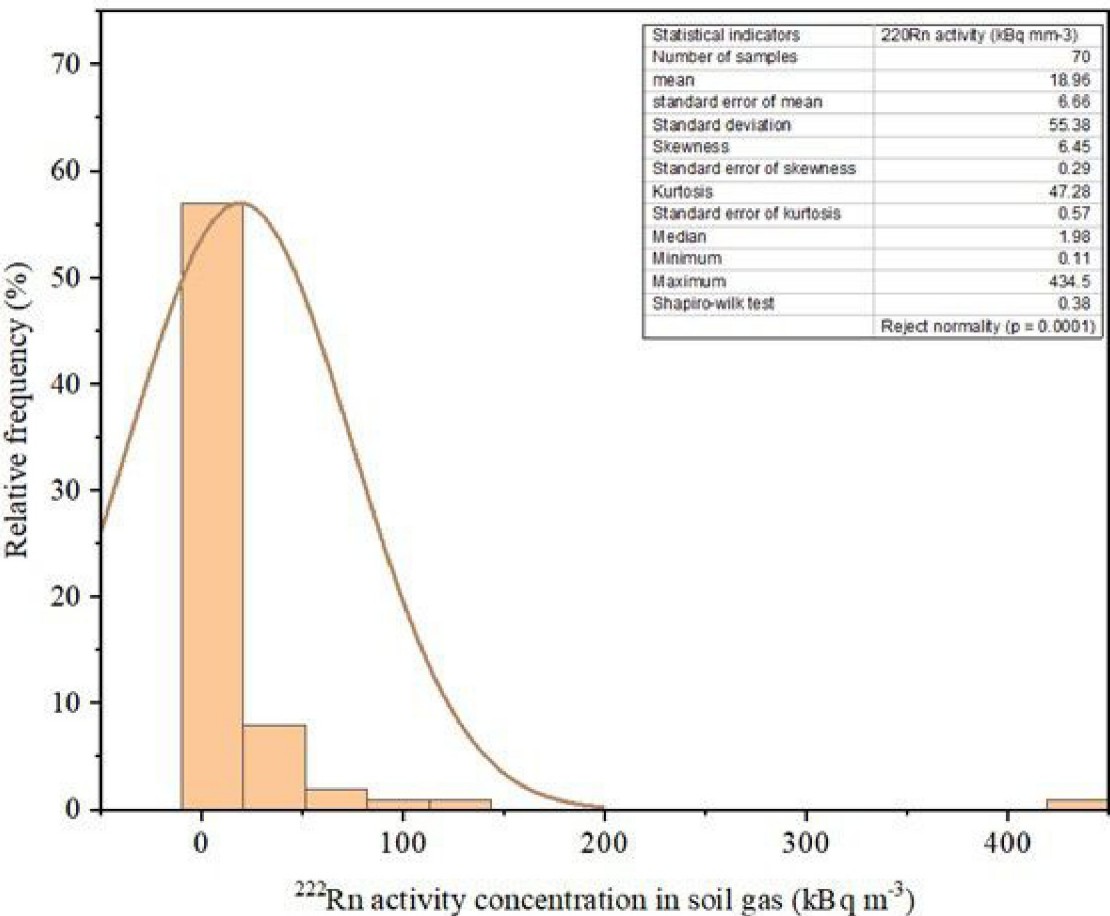

| Statistical indicators | 220Rn activity (kBq mm-3) |
|---|---|
| Number of samples | 70 |
| mean | 18.96 |
| standard error of mean | 6.66 |
| Standard deviation | 55.38 |
| Skewness | 6.45 |
| Standard error of skewness | 0.29 |
| Kurtosis | 47.28 |
| Standard error of kurtosis | 0.57 |
| Median | 1.98 |
| Minimum | 0.11 |
| Maximum | 434.5 |
| Shapiro-wilk test | 0.38 |
| | Reject normality (p = 0.0001) |

**Fig 2. Histogram of radon activity concentration distribution in soil and statistical indicators.**

The radon activity concentration map computed by Kriging/Cokriging interpolation is elucidated in Fig 3. The class breaks were determined using geometrical interval and a nine-color code of the standard model in ascending order with value labels from low to high activity concentration. Elevated radon activity was found in soils derived from granite geological formation and recent riverine alluvium, with mean values ranging from 4.28 to 44.48 kBq m$^{-3}$ and a maximum of 434.5 kBq m$^{-3}$ which is located at the most western granitic region of the state. The soil types include Rengam (Paleudult), Bukit Temiang (Hapludult), Akob (Aeric Tropaquept), Typic Sogomana (Kandiaquult), and Holyrood (Kandiadult). The area is located on the main granite range of Peninsular Malaysia, which consists of igneous granite rock. High radon concentrations are associated with granite rocks, therefore high radon activity concentration in soils derived from such bedrock was expected, as duly obtained. Moderate mean radon activity concentrations were found in miscellaneous soils (Urban and Mined land), and soils derived from sedimentary/metamorphic rocks Serdang (Paleudult-Hapludox), Chenian (Kanhapludlt) which ranged from 8.48 to 9.62 kBq m$^{-3}$. The lowest mean radon activity concentration (4.28 kBq m$^{-3}$) was found in marine soils (soils on coastal plains), whose parent materials are of marine, estuarine clays, and organic deposits, with the highest value of (27.6 kBq m$^{-3}$). These soil types consist of, Kranji (Sulfaquent), Selangor (Aeric Tropic Fluvaquent), Briah (Tropic Fluvaquent), and Rudua (Spodic Quartzipsammant).

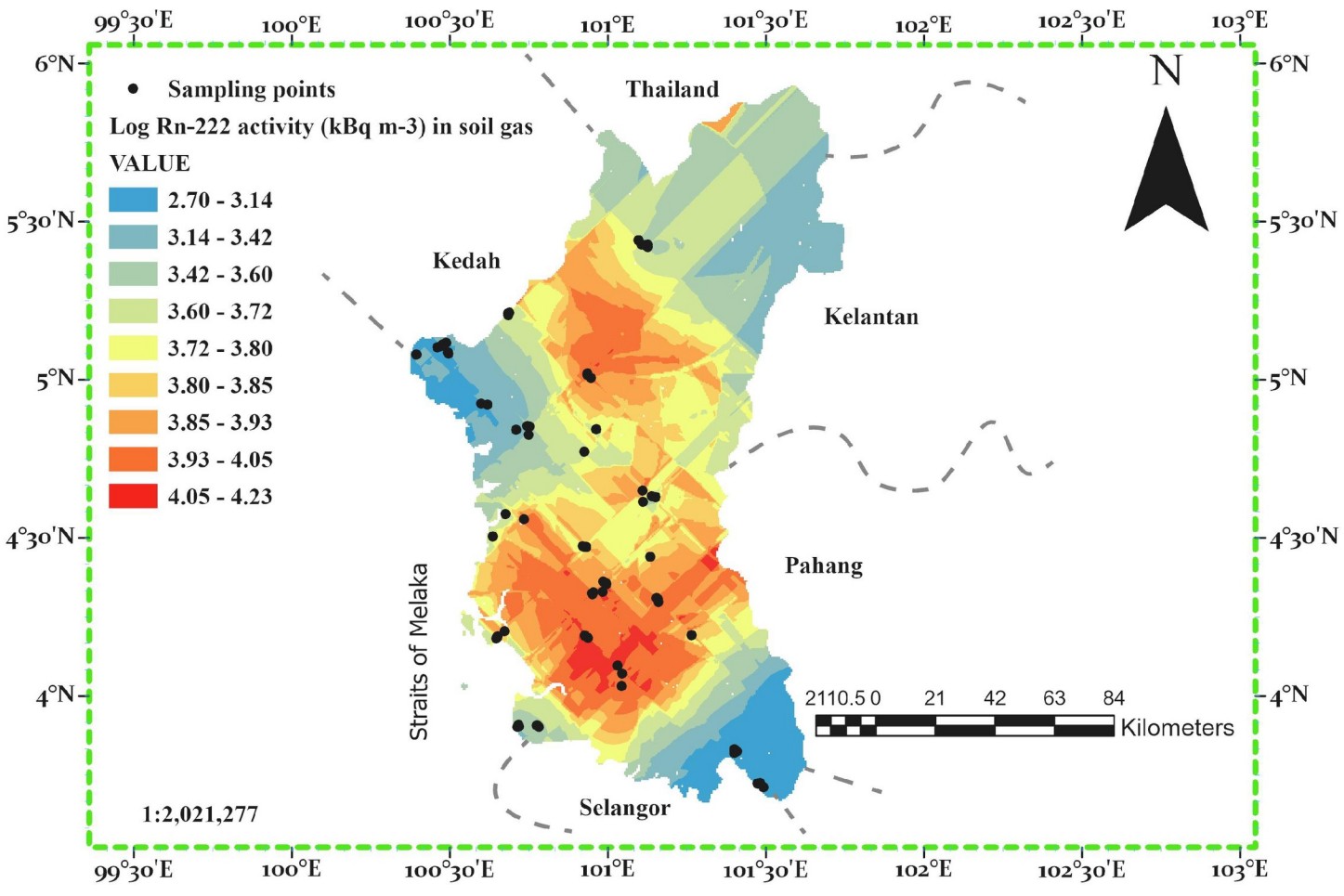

**Fig 3. Map of radon activity concentration in soil gas Perak state.**

## Radon activity concentration for each geological type

Presented in Fig 4 is the box plot of the radon concentration according to the geological formations of the study area. The sampling points were allocated to each geological code using the ArcGIS Pro software. The number of samples in each classification were Quaternary (22), Triassic-Jurassic (26), Silurian (10), and Intrusive rocks (12). The overall median (1.98kBq m$^{-3}$) value of $^{222}$Rn activity concentration for this study is higher than the median values obtained from, the areas under the Quaternary (1.13 kBq m$^{-3}$) and Silurian (1.4 kBq m$^{-3}$) geological formations. The Triassic-Jurassic geological formation has a median value of 1.97 kBq m$^{-3}$ almost the same as the overall median and data range of 1.0 to 106.5 kBq m$^{-3}$. The highest median (10.43 kBq m$^{-3}$) is from regions with geological formations that composes of undifferentiated intrusive igneous rocks of granitic origin (Intrusive rocks). The data range from the intrusive rock regions ranged from 9.81 to 434.5 kBq m$^{-3}$. Thus it conforms to the UNSCEAR (2006), in which it is stated that a high concentration of natural radionuclides is found in rocks of granitic origin.

The mean $^{222}$Rn activity concentration values in soil gas for each geological formation are also illustrated in Fig 4 The overall mean (18.96 kBq m$^{-3}$) is higher than the means obtained from Quaternary (5.32 kBq m$^{-3}$), Triassic-Jurassic (17.02 kBq m$^{-3}$), and Silurian (5.16 kBq m$^{-3}$) geological formations respectively. The highest mean (60.36 kBq m$^{-3}$) is obtained from

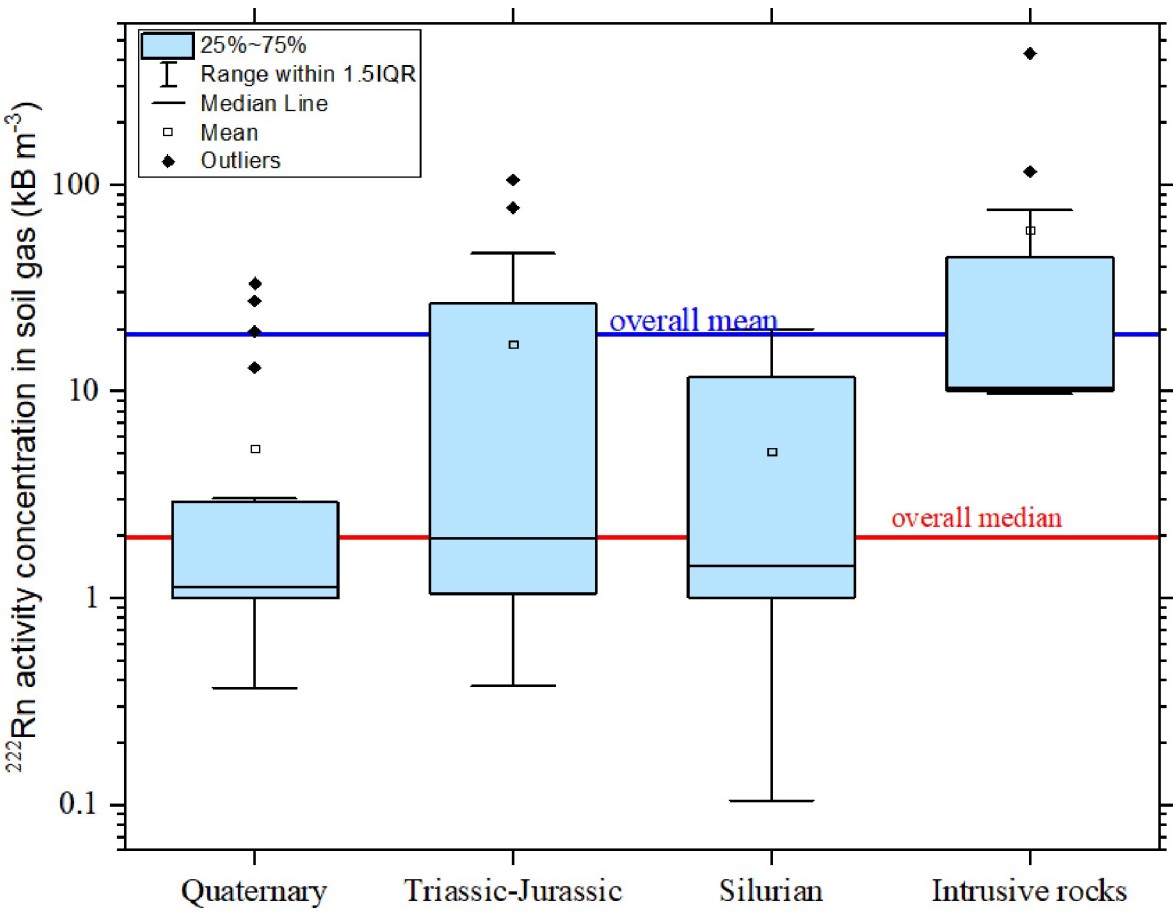

**Fig 4. Box plot of radon activity concentration in geological formations of Perak state.**

the geological formations that compose of undifferentiated intrusive igneous rocks of granitic origin (Intrusive rocks).

## Soil gas permeability

As earlier indicated, soil permeability (k) is a basic parameter that is paramount in the release of radon to the atmosphere or its accumulation in the soil [49]. High permeability allows the upward migration of radon, enabling its' exhalation to the atmosphere, while low permeability limits the radon exhalation rate to the atmosphere. Soil permeability range has been grouped into three classes (High, Medium, and Low) in this work, following the threshold established by Neznal, Neznal [23] in Table 1.

A summary of the data distribution of the soil permeability for the study area is displayed in the box plot in Fig 5. To make the measurement condition near homogenous, the soil permeability data were collected during a season of low rainfall. The soil permeability of the study falls mostly in

**Table 1. Soil gas permeability classifications [23].**

| Classification | Permeability (m$^2$) | Number of samples |
|---|---|---|
| High | k> $4.0 \times 10^{-12}$ | 14 |
| Medium | $4.0 \times 10^{-12} \geq$ k $\geq 4.0 \times 10^{-13}$ | 20 |
| Low | k< $4.0 \times 10^{-13}$ | 36 |

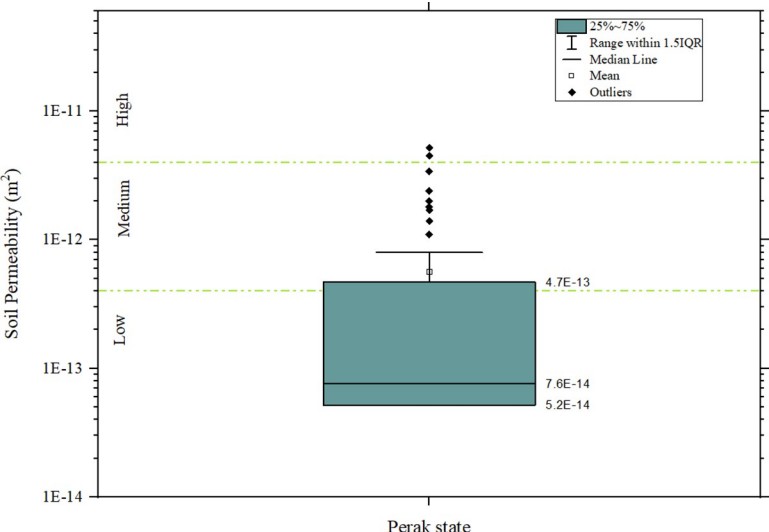

**Fig 5. Box plot of soil gas permeability of Perak state.**

the low-medium range, as can be observed in the box plot. The high permeability is about 3% of the distribution. Most of the data was taken from low to medium permeability areas.

The soil gas permeability distribution according to geological formations is shown in Fig 6. The Overall median ($7.6 \times 10^{-14}$ m$^2$) of soil gas permeability was lower in all except the Quaternary ($5.20 \times 10^{-14}$ m$^2$) geological formation. The highest median value was obtained from the geological formations compose of undifferentiated intrusive igneous rocks of granitic origin (Intrusive rocks).

Fig 6 displays the mean soil permeability across the geological formations as well. The highest mean ($7.12 \times 10^{-13}$ m$^2$) value was obtained from the geological formations composed of undifferentiated intrusive igneous rocks of granitic origin (Intrusive rocks). The mean of the soil gas permeability in the Triassic-Jurassic (Sedimentary rocks with conglomerates) ($6.32 \times 10^{-13}$ m$^2$) and Intrusive rocks ($7.12 \times 10^{-13}$ m$^2$) were higher than the overall mean respectively. However, the means of soil permeability from the Silurian ($5.06 \times 10^{-13}$ m$^2$) and Quaternary ($3.11 \times 10^{-13}$ m$^2$) geological formations were lower than the overall mean. The soil gas permeability in the Quaternary was generally lower. This could be because of its proximity to coastal areas by the straits of Melaka.

The soil gas permeability map is displayed in Fig 7. The area with low permeability were mostly located in the west of Perak in the coastal lands (mostly marine and riverine soils). The medium soil gas permeability was in the central, northern, and southern parts of Perak state, which constitute mostly soils from sedimentary and metamorphic rocks. The high permeability was in the eastern part of Perak state, which is mostly soils from granite rocks. Although efforts were put to take measurements on undisturbed lands, there were certain urban areas in which it was difficult to achieve that. In general, there were isolated areas of medium permeability in high permeability areas and low permeability in medium permeability areas and vice-versa.

### Geogenic radon potential (GRP) and radon prone areas

The GRP was calculated based on the formula proposed by Neznal, Neznal [23] defined in Eq (2).

$$GRP = \frac{C_{Rn}}{-\log(k) - 10} \tag{2}$$

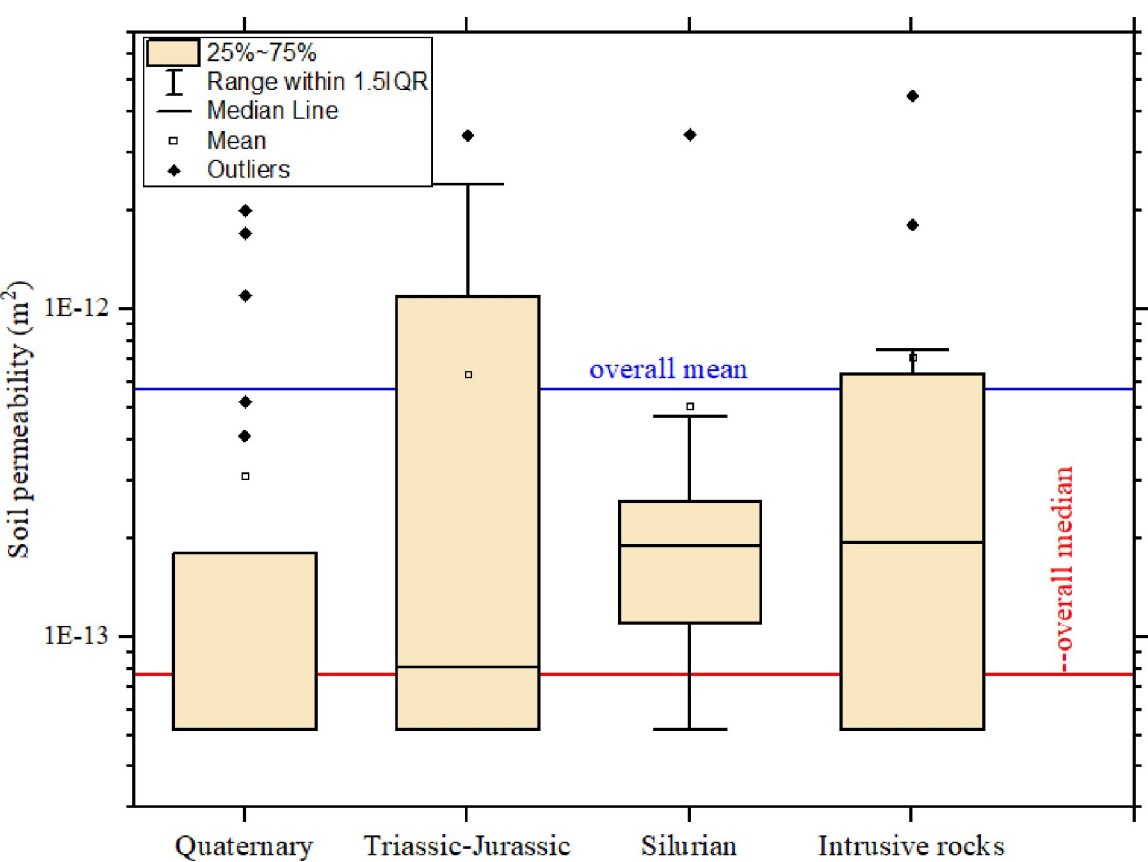

**Fig 6. Box plot of soil gas permeability in geological units.**

where $C_{Rn}$ (kBq m$^{-3}$) is the $^{222}$Rn concentration in soil gas and k (m$^2$) is the permeability of the soil. Higher mean GRP values imply greater potentials for radon to migrate through the soil [16].

Radon-prone areas are usually categorized using the Radon Index (RI) based on the values of GRP. Many years of extensive research in the Czech Republic has led to categorizing radon prone areas into three, depending on the GRP value: low RI (GRP < 10), medium RI (10 < GRP < 35), and high RI (GRP > 35) [2, 22–24].

Using the Czech classification, the GRP values of this study are displayed in Fig 8. The areas with high RI (GRP >35) [6%] are in granitic regions. The areas with medium RI (10 < GRP > 35) [14%] are in Triassic-Jurassic regions, with isolated cases in quaternary. The bulk (80%) of the study locations are in the region with low RI (GRP < 10). An area might be categorized as having low RI, but it is possible to have some isolated radon concentration above the reference level in such areas. Similarly, an area categorized as having high RI can also have some isolated radon levels below the reference level. Processes in the soil causing such low/high concentrations could occur as a result of several factors such as soil permeability, particle size, porosity, water content near saturation level, soil comprising of clay, exhalation, advection, temperature, and diffusion [2].

The GRP map produced using Kriging\cokriging interpolation is presented in Fig 9. A three-color code was used in categorizing the RI. The map indicates that a large part of the state can be categorized in the low RI areas. The regions with medium to high RI are mostly concentrated in the central and north-western parts of the state, which is predominantly made up of soils derived from sedimentary/metamorphic rocks, granite rocks, and miscellaneous

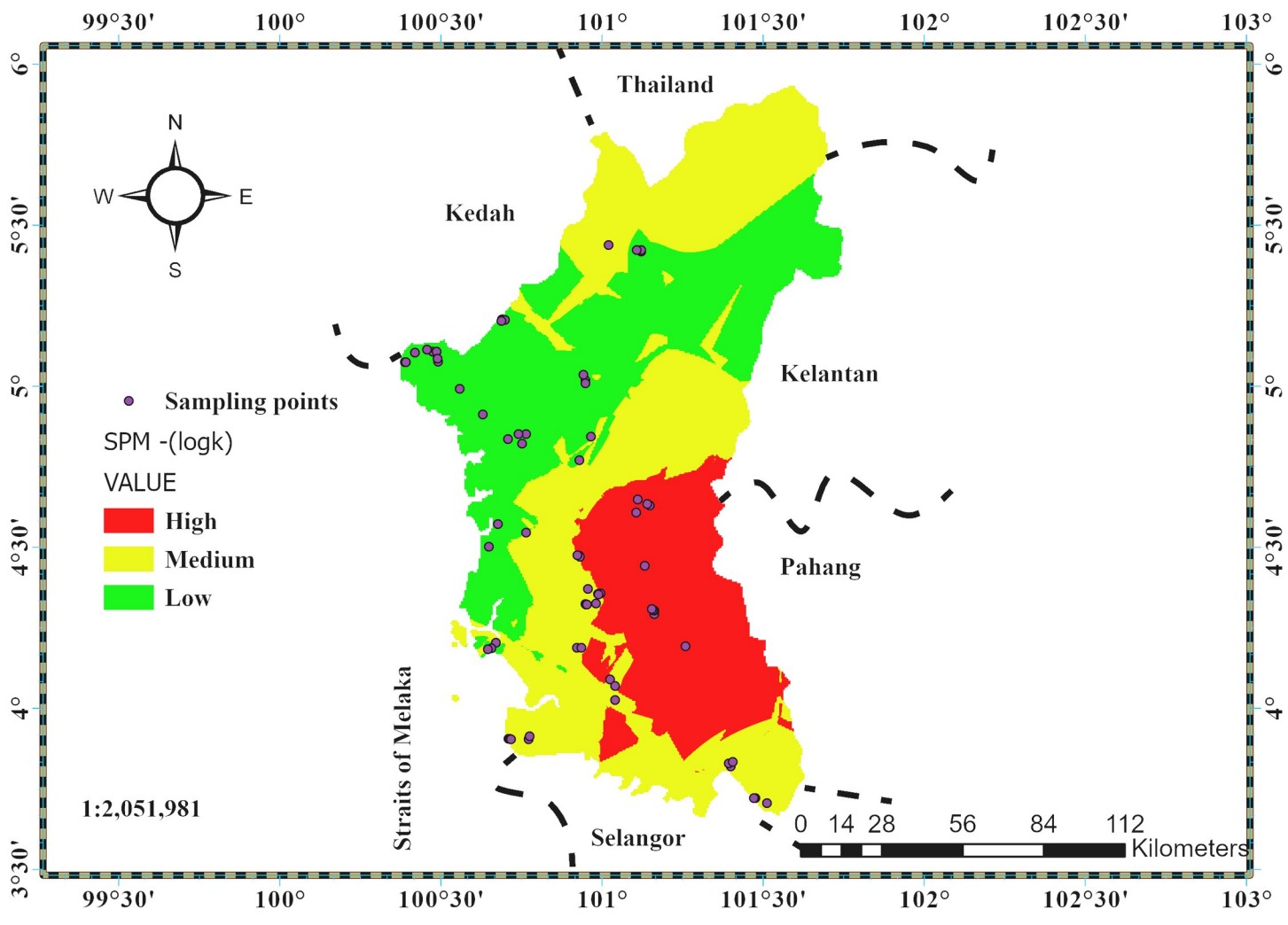

**Fig 7. Soil permeability map of Perak state.**

(urban and mined) soils. Buildings in such areas are prone to indoor radon gas accumulation, which is affected by the environmental conditions (temperature gradient) that influence the radon gas transport, to be either by advective transport (temperature gradient present) as the dominant mechanism or transport by diffusion (temperature gradient absent) as the dominant mechanism. Displayed in Fig 10 is the empirical semi-variogram in the theoretical stable model. Most of the empirical semivariances fall along the model line as required.

The cross-validation procedure that was used in this work is the "leave one out" method. It involves removing a single point in the data set and using all remaining points to predict the location of the point removed. The predicted value is then compared to the measured value and many statistics are generated to determine the accuracy of the prediction. The model cross-validation results in Table 2 indicate that the predicted GRP values match the soil gas radon values measured. The continuous ranked probability score (CRPS) in Table 2 represents the diagnostic score which measures each observed data's deviation from the predictive cumulative distribution function. The mean error of prediction (ME) and the mean Standardized error (MSE) are 0.01 and 0.005 respectively and near zero, suggesting that the interpolation technique used is hinged on true values and the model is a particularly good one. The average

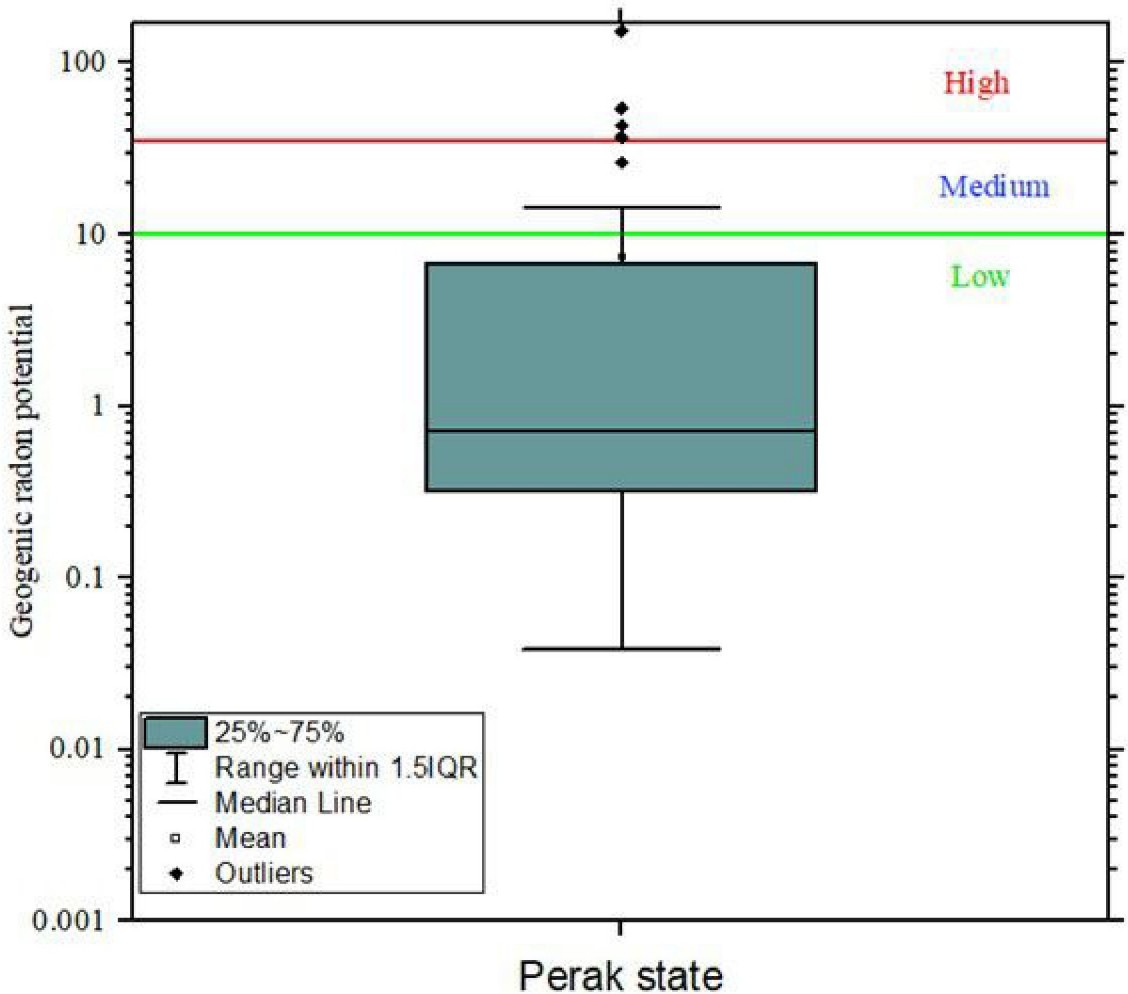

**Fig 8. Box plot of GRP classification Perak state.**

standard error (SE) is 15.44 and is less than the root-mean-square error (RMSE) of prediction, which is 18.23, indicating a slight underestimation of the variability in predictions by the interpolation method.

Displayed in Fig 11 is the standard error of the prediction map. The map shows that near the sampling points the prediction is good. However, as we move away from the sampling points the standard error increases. In the areas of high GRP, the uncertainty is relatively moderate ($< 20$). In low GRP regions, the uncertainty is relatively low ($< 16$).

## Conclusion

Pioneer maps of radon prone areas in Perak state, Malaysia were produced. The regions with high radon risk areas are mostly concentrated in the central and southern part of the state, which is predominantly made up of soils derived from granite rocks. Based on the GRP classification low $^{222}$Rn values are observed in the south-eastern and northern-central sectors of the study area which is underlined with medium soil permeability. From the radon risk index based on the GRP classification of Perak state, the area is mostly at low risk. However, there are areas that are in the medium radon risk, with isolated high GRP values. It should be noted

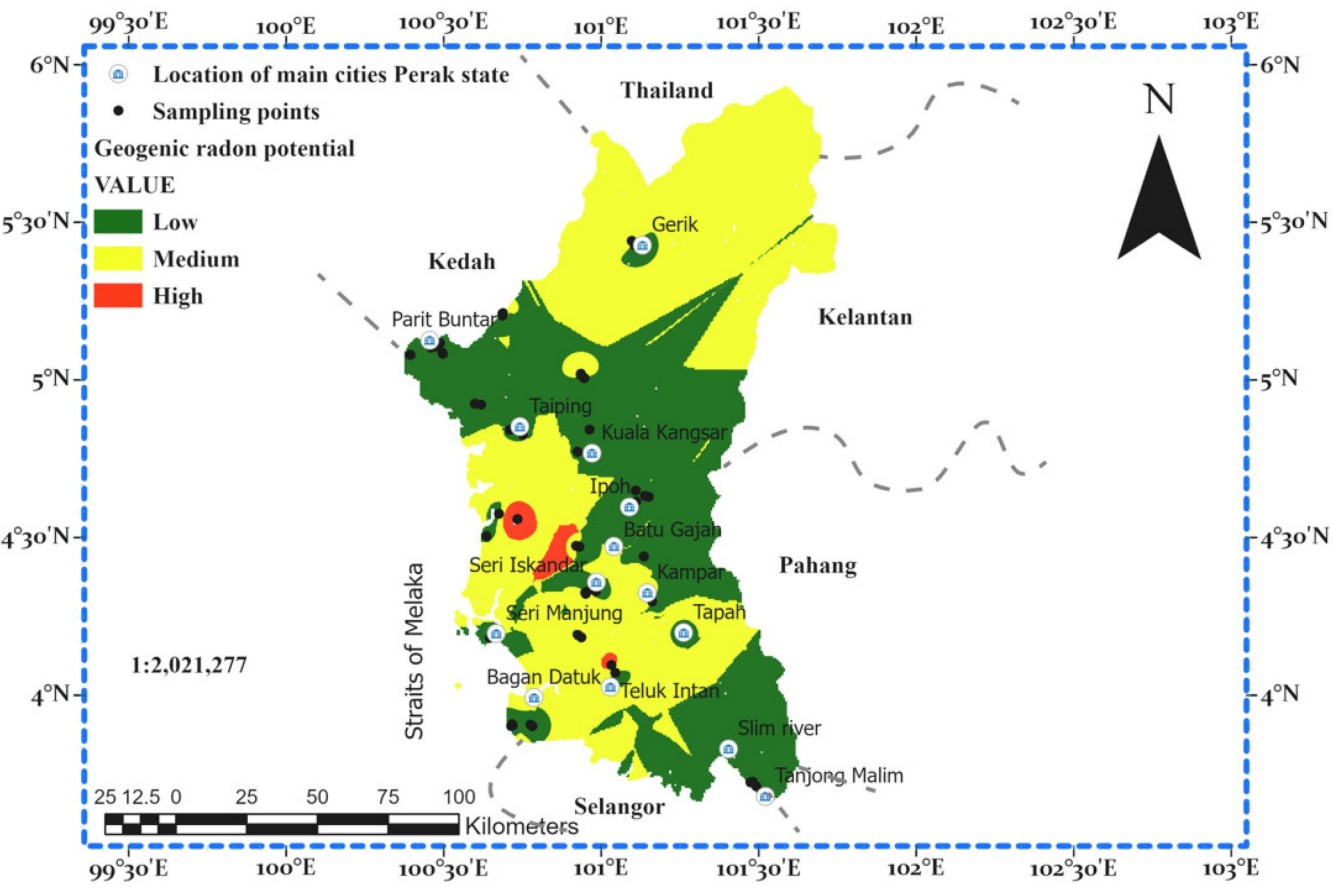

**Fig 9. Geogenic radon potential map of Perak state Malaysia.**

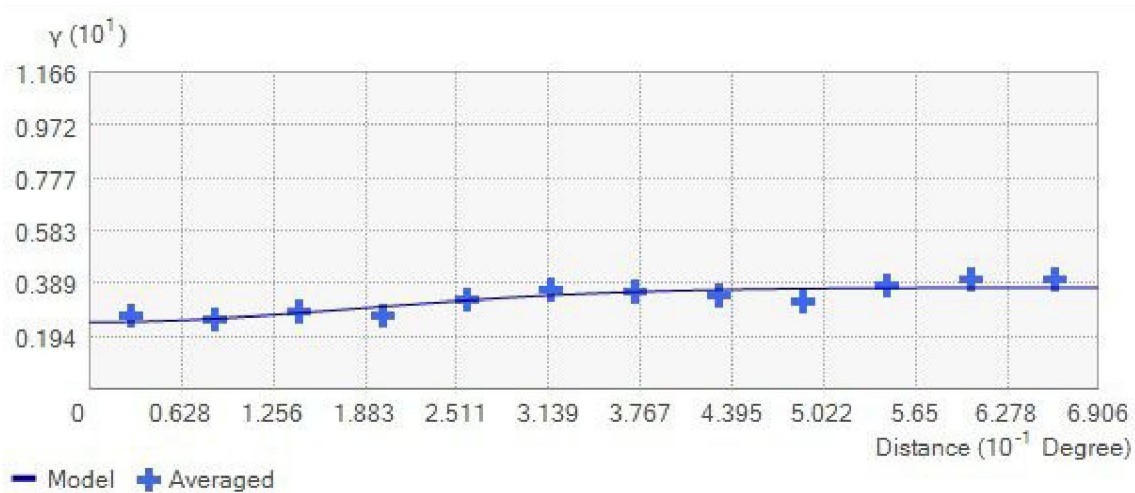

**Fig 10. Semi-variogram.**

**Table 2. Cross-validation results for model.**

| Parameter | Value |
|---|---|
| Average CRPS | 7.60 |
| Inside 90 percent interval | 92.86 |
| Inside 95 percent interval | 94.24 |
| Mean error | 0.01 |
| Root-mean-square error | 18.23 |
| Mean standardized error | 0.005 |
| Root-mean square standardized | 1.07 |
| Average standard error | 15.44 |

that although an area might be categorized as low risk, it is possible to have radon concentration above the reference level. Similarly, an area levelled as high risk can have radon levels below the reference level.

The authorities saddled with the responsibility of radon monitoring and control will find the maps produced in this study a valuable tool in the prevention and mitigation of radon in dwellings and workplaces. It could also help in the inclusion of building techniques that are radon resistant.

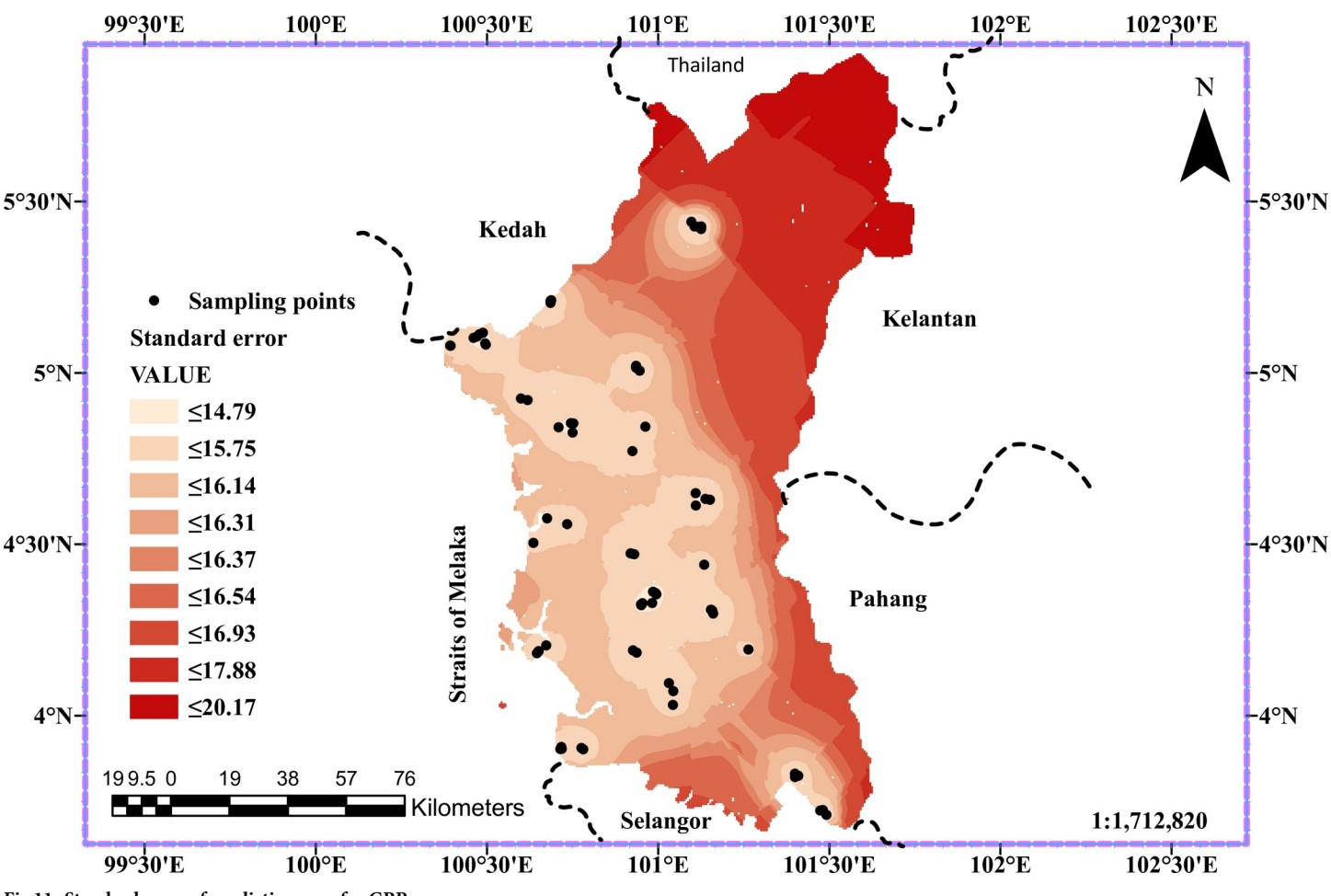

**Fig 11. Standard error of prediction map for GRP.**

## Supporting information

**S1 File. Minimal data set.**
(DOCX)

## Acknowledgments

We would like to express our special gratitude to the Department of Mineral and Geoscience Malaysia under the Ministry of Energy and Natural Resources for approving our request to use the Geological Maps of Perak.

## Author Contributions

**Conceptualization:** Habila Nuhu, Suhairul Hashim, Muneer Aziz Saleh, Mohamad Hidayat Jamal, Rini Asnida Abdullah, Sitti Asmah Hassan.

**Formal analysis:** Habila Nuhu.

**Funding acquisition:** Suhairul Hashim, Muneer Aziz Saleh, Mohamad Hidayat Jamal, Rini Asnida Abdullah, Sitti Asmah Hassan.

**Investigation:** Mohamad Syazwan Mohd Sanusi.

**Methodology:** Habila Nuhu, Muneer Aziz Saleh, Mohamad Syazwan Mohd Sanusi.

**Supervision:** Suhairul Hashim, Muneer Aziz Saleh.

**Validation:** Mohamad Syazwan Mohd Sanusi, Ahmad Hussein Alomari.

**Writing – original draft:** Habila Nuhu.

**Writing – review & editing:** Habila Nuhu, Suhairul Hashim, Ahmad Hussein Alomari, Mohamad Hidayat Jamal, Rini Asnida Abdullah, Sitti Asmah Hassan.

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
