## [Decision Letter · Decision Letter 0]

19 Nov 2020

PONE-D-20-31278

Soil gas radon and soil permeability assessment: Mapping radon risk areas in Perak State, Malaysia.

PLOS ONE

Dear Dr. Nuhu,

Thank you for submitting your manuscript to PLOS ONE. After careful consideration, we feel that it has merit but does not fully meet PLOS ONE’s publication criteria as it currently stands. Therefore, we invite you to submit a revised version of the manuscript that addresses the points raised during the review process.

The two reviewers have concluded that this work is worth publishing but that it requires major revision first. Both reviewers have made constructive comments to improve the paper. Please address these and re-submit.

We look forward to receiving your revised manuscript.

Kind regards,

Elisabeth Bui

Academic Editor

PLOS ONE

Journal Requirements:

2. In your Methods section, please provide additional information regarding the permits you obtained to collect samples for the present study. Please ensure you have included the full name of the authority that approved the field site access and, if no permits were required, a brief statement explaining why.

3. We note that Figures 1, 3, 7, and 8 in your submission contain map images which may be copyrighted. All PLOS content is published under the Creative Commons Attribution License (CC BY 4.0), which means that the manuscript, images, and Supporting Information files will be freely available online, and any third party is permitted to access, download, copy, distribute, and use these materials in any way, even commercially, with proper attribution. For these reasons, we cannot publish previously copyrighted maps or satellite images created using proprietary data, such as Google software (Google Maps, Street View, and Earth). For more information, see our copyright guidelines: http://journals.plos.org/plosone/s/licenses-and-copyright.

(1) You may seek permission from the original copyright holder of Figures 1, 3, 7, and 8 to publish the content specifically under the CC BY 4.0 license. 

Reviewers' comments:

Reviewer's Responses to Questions

**Comments to the Author**

1. Is the manuscript technically sound, and do the data support the conclusions?

Reviewer #1: No

Reviewer #2: Partly

2. Has the statistical analysis been performed appropriately and rigorously? 

Reviewer #1: No

Reviewer #2: Yes

3. Have the authors made all data underlying the findings in their manuscript fully available?

Reviewer #1: No

Reviewer #2: Yes

4. Is the manuscript presented in an intelligible fashion and written in standard English?

Reviewer #1: No

Reviewer #2: Yes

5. Review Comments to the Author

Reviewer #1: The manuscript "soil gas radon and soil permeability assessment: mapping radon risk areas in Perak State, Malaysia" by Nuhu et al. presents a case study in a region where data and maps of the radon related risk have been rarely published. Therefore, I generally encourage publication of this pioneering work. However, before publication I suggest the authors to substantially revise the manuscript, especially regarding these general aspects:

1. A large fraction of the observed data on soil gas radon concentration seem to be extraordinarily low (more in the range of outdoor air rather than soil gas). Please give a physical reasoning for these large amount of observations with very low values.

2. The interpolation method (empirical bayesian kriging) is not described at all. Please give a brief description. Also, please elaborate why you have chosen this interpolation technique (which is rather novel and sophisticated) and not a simpler one such as a more basic kriging alternatives. What are the advantages of EBK?

3. How was cross-validation conducted? Please give details on fold assignment. Did you consider spatial autocorrelation in the observational data during cross-validation? What is the range of spatial auto-correlation in your data?

4. You showed a dependence of GRP on geology. Why do you not exploit this knowledge for spatial prediction/interpolation, e.g. by using geology as co-variable?

5. The conclusion is only partly supported by your results. The high risk of granitic areas seems to be rather your hypothesis than your finding. Large parts of the high GRP areas are not in the granitic areas. For instance, a large fraction of the area covered with quaternary sediments (in the south and south-west of the study area) has a higher GRP than the granitic areas. This might be a consequence of the interpolation technique that you selected by not implementing geology as co-variable.

6. In many cases the language is not precise enough, partly ambiguous and grammatically not always sound.

Minor comments

• L50: Rn is one of the leading causes for lung cancer, not for cancer in general

• L54: “generate higher indoor Rn levels”

• L 56: these methods are not geology-based. I guess you mean statistical methods (geostatistics, regression) using co-variable/proxy data

• L 57: Ariel? Please explain

• L76: What do you mean with “baseline” data? Unclear.

• L 93: consider replacing geological “contexts” with “regions”

• L96-98: The description of the sampling design is unclear to me. Please specify. Did you use the same fraction of measurements per geological region or did you sample preferentially in areas with expected higher values or higher geological heterogeneity?

• Fig.1: What is the scale of the map? Please add sampling locations. Your geological classification in the legend is a mix of stratigraphic and lithologic classes. Please be consistent. Either stratigraphic or lithologic or lithostratigraphic.

• Please indicate sampling locations also in Fig. 3, 7 & 8. This will help interpreting the maps.

• Fig 2: please consider using smaller bins (e.g. 5 or 10 kBq/m³) for the histogram. In the box statistical indicators, a median of 0.15 seems to be unrealistic given my experience and also the data shown in the plot. Consider using standard deviation instead of error since you want to give a measure of dispersion in your data set rather than an error/uncertainty of an individual observation. There is no boxplot in this figure (figure caption)

• I would recommend not using two decimal digits because it pretends an accuracy that is not existing in reality

• L132: A soil gas Rn concentration of 0.01 kBq/m³ is a very low value. This is rather a Rn concentration observed in outdoor air than in soil (where Rn is continouosly produced). I would suspect such low values as erroneous measurements (e.g., due to contamination of your measured volume with outdoor air). If you consider these values to be realistic please elaborate on the processes in soil causing such low concentrations.

• L136: You performed log-transformation of your data prior to geostatistical modelling. How did you backtransform your data to the original scale after interpolation.

• L 166: “marine soils” are misleading for soils with marine sediments as parent material. Please consider revising.

• Fig. 3. In this map a few apparent artefacts (e.g., in the east next to the label of Kelantan) are existent. Further, I suggest using even value breaks instead of giving two decimal digits.

• L 174ff: I suggest shifting the explanation of a boxplot into the caption of Fig4 because it breaks somewhat the flow in your text and it can be assumed that most readers are familiar with the concept of a boxplot.

• L180: The highest mean Rn conc. was found in Granite but the highest median in sed./metamorph. Please be more precise.

• Fig 4: Does Fig. show Rn data on the original scale or log-transformed data. If it shows data on the original scale, most values are exceptionally low which would require a physical explanation. How many samples do you have in each class? Please give n per class.

• L 211ff: the GRP is practically dimensionless since the log of k is used in eq. 1

• L227: Please describe your geostatistical model. What are the parameters of your variogram?

• L227: What is the reasoning for using 9 colour codes? If you do a classification in low, medium and high, it would be straightforward to use only 3 codes

• L240: Please describe your cross-validation procedure. How did you assign observation to the testing fold? Random, stratified, spatially blocked?

• Fig. 8: resolution of the figure is poor.

• L259ff: your conclusion is not fully supported by your results. You claim that high GRP areas are mainly found in the granitic areas. In fact, the high GRP regions in Fig 7 and the outline of the granitic rocks (Fig1) do overlap only partly. I feel that this granitic regions being at higher risk is rather your hypothesis than your finding. Further, large parts of the high GRP areas are located in the quaternary sediments. This might be a consequence of their proximity to high GRP observations in the nearby granitic areas. If this is the case (I can only speculate on this issue because information on sampling locations is missing), I suggest considering the implementation of your knowledge (dependence of GRP on geology) in your spatial prediction/interpolation model (e.g., by using geology as co-variable).

• L 267: Why do you refer to a reference level of 300 Bq/m³? Your paper deals with the GRP, while the reference level of 300 Bq/m³ (or 100 Bq/m³or 200 Bq/m³) applies to indoor environments. The indoor reference level cannot be applied to your GRP estimates.

Reviewer #2: Dear Dr. Bui

The manuscript by Nuhu et al. (Ref. No.: PONE-D-20-31278) deals an important issue on study of spatial distribution of geogenic radon potential. The experimental part of the study includes measurements of radon concentration in soil gas and soil permeability; the applied geostatistical method is consistent with the scope of the study. However, the geostatistical analyses are not described in sufficient detail, and the discussion is quite superficial therefore, in my opinion, a MAJOR revision of the manuscript is necessary.

My detailed comments on manuscript are given below:

Introduction

Lines 53-54: Please, check the definition of RPA, and clarify its relationship with indoor radon. “Radon Prone Area” is a term that describes an area where the probability of enhanced radon concentration is much higher than average. The establishment of such areas, by national authorities, can be based on geological conditions and/or the results of measuring campaigns of indoor radon concentration. See ICRP 1993 (Protection against radon-222 at home and at work. ICRP Publication 65. Paragraph 3.3)

Line 57: “Ariel” ?

Lines 67-68: High GRP indicates a high availability of geogenic radon that can enters indoor. Also, express clearly that GRP is related to Ra content (but also U content!) because it is a Rn source, whereas permeability (but also faults!) is related to Rn mobility.

Line 69: I suggest to add here a short review of the methods previously used to map GRP.

Lines 75-76: Please, summarize the results of previous studies.

Line 78: What data? Please, add description and reference.

Line 82: A strict definition of “radon risk” should be presented.

2.0 Materials and Method

In this section, after the description of the study area and the analytical techniques, I expected to read a paragraph about the geostatistical technique used to map the GRP. Practically, the discussion is based on the maps elaborated using Empirical Bayesian Kriging (EBK), but the method description is absent. Please, add the description of EBK model and explain why you chose this interpolation method. What semivariogram did you select? What the maximum number of points in each local model? What software did you use? What is the resolution of your maps?

2.1 Study area and sampling points

Lines 92-95: The geological description is too scarce, please add more information. Faults and structural discontinuities may be important pathways for radon movement in the subsoil, so you should describe the tectonic frame of the study area.

Line 96-97: please, add the location of sampling points in Fig.1. Also, calculate the sampling density. How many samples for each geological formation?

2.2 Measurement of radon in soil gas

Please, specify in what period of the year and meteorological conditions the measurements were performed.

Line 109: Did you use a drying tube to maintain the relative humidity at low values?

Results and discussion

3.1 Activity concentration of radon in soil gas

Lines 131-132: for non-normal distribution, better to use geometric mean

Lines 132-134: this description is does not fit to the histogram of fig. 2, probably a histogram with more classes will be better describe your data. Please, consider using a normal probability plot; it can help to recognize both anomalous samples and multiple populations in log-normally distributed data

Lines 152-153: here the Authors introduce for the first time the Empirical Bayesian Kriging; see the comment above.

Lines 153-155: this description is unnecessary; better to specify how you have determined the class breaks (e.g., manual interval, quantile, natural breaks etc.)

Line 157: “ Manjung District”, and other localities cited later in the text (e.g., line 214), are unknown to non-Malaysian. Please, add it in the Figures.

Lines 158-159: An explanation is needed of how this list of soil types is used and how these soils were identified. Did you identified these soils in field? Did you extrapolated from a pedological map? Is there a correlation between soil types and Rn concentration and/or permeability?

Lines 159-162: This statement causes some doubts. The Authors identified the area with the highest values of soil gas radon in Manjung District, which is located in the southwestern part of the state of Perak; here, granite rocks outcrops and therefore the Authors explain the high Rn values because it derived from such bedrock. However, the geological map of fig. 2 shows that granite rocks extensively outcrop in the eastern part of the study area, where the soil gas Rn concentrations are not particularly high (Fig.3). So, how the Authors explain these spatial distribution? Did you consider the presence of permeable structures such as fractures and/or faults?

Line 163: What mines/quarries are in the study area? Where they are located?

Line 169: I suggest to add a table (or a box-plot) with the statistics (average values, ranges of values) of soil gas Rn concentration and permeability in different soil types, I think that it can be help the reader.

3.2 Radon activity concentration for each geological type

Sincerely, there is no reason to separate this paragraph from the previous one.

Lines 174-178: the general description of box-plot should be moved in the method section. A sub-paragraph describing statistical and geostatistical elaborations might be appropriate.

3.3 Soil gas permeability

Lines 197-198: Please, add a map of spatial distribution of permeability, it could be very useful here. Please, also discuss how permeability influences radon in soil, in general and in your case study.

Table 1: Add the number of your samples that fall within each class of permeability.

3.4 Geogenic Radon Potential (GRP) and Radon prone areas

Line 210: “Many years of extensive research in the Czech Republic”…please, add references.

Line 232: the title of the paragraph is “Geogenic Radon Potential (GRP) and Radon prone areas”, but where in the text the RPA are described?

Please, compare the map of GRP with the map of Terrestrial Gamma Dose Rate (TGDR) published by Ramli et al (2016) - Predicting terrestrial gamma dose rate based on geological and soil information: case study of Perak state, Malaysia. J. Radiol. Prot. 36(1), 20-36.

4.0 Conclusion

Lines 268-270: this concept should be emphasized in Discussion.

Figures

Figure 1: add the location of sampling points, the prevailing tectonic lines, names of the district cited in the text

Figure 2: delete “and box plot” in the figure caption

Figures 3 and 7: add the corresponding semivariograms.

Figure 4: what represent the blue dashed lines? How many samples for each geological type?

Figure 5: add box plots classified according to geological formations and/or soil types.

Figure 7: add a layer of urbanized areas, in order to evaluate if the areas of high RI are already urbanized or not yet; in the first case, the map could indicate the areas where it could be necessary monitoring the indoor radon concentration, in the second case the map could be a tool for future urbanistic plans.

6. PLOS authors have the option to publish the peer review history of their article (what does this mean?). If published, this will include your full peer review and any attached files.

Reviewer #1: No

Reviewer #2: No

---

## [Author Response · Author response to Decision Letter 0]

15 Jan 2021

Manuscript Number: PONE-D-20-31278

Response to reviewers’ comment

Academic editor

 Authors reply:

Thanks for your suggestion, please note that we have revised the manuscript format accordingly. 

2. In your Methods section, please provide additional information regarding the permits you obtained to collect samples for the present study. Please ensure you have included the full name of the authority that approved the field site access and, if no permits were required, a brief statement explaining why.

 Authors reply:

Thanks for your suggestion, please note that there is no permit required for the on-ground radon survey as the sampling locations are located or selected in personal territory areas or highly risk areas. On top of that, this is government funding work and special privilege is granted for the researcher. A brief statement is given in methodology to address these comments. 

3. We note that Figures 1, 3, 7, and 8 in your submission contain map images which may be copyrighted. All PLOS content is published under the Creative Commons Attribution License (CC BY 4.0), which means that the manuscript, images, and Supporting Information files will be freely available online, and any third party is permitted to access, download, copy, distribute, and use these materials in any way, even commercially, with proper attribution. For these reasons, we cannot publish previously copyrighted maps or satellite images created using proprietary data, such as Google software (Google Maps, Street View, and Earth). For more information, see our copyright guidelines: http://journals.plos.org/plosone/s/licenses-and-copyright.

(1) You may seek permission from the original copyright holder of Figures 1, 3, 7, and 8 to publish the content specifically under the CC BY 4.0 license. 

 Authors reply:

Thanks for your suggestion, please note that all Fig. 1,3,7,8 are original work from authors, digitised images which produced from ArcGIS software. On top of that the geological map of Peninsular Malaysia is free for educational and research purpose. Please refer to our communication email with the officer at the Department of Mineral and Geoscience Malaysia under the Ministry of Energy and Natural Resources for approving our request to use the Geological Maps of Perak. The acknowledgement to this department has been added to express our gratitude on their cooperation.

Reviewer #1 

1. A large fraction of the observed data on soil gas radon concentration seem to be extraordinarily low (more in the range of outdoor air rather than soil gas). Please give a physical reasoning for these large amount of observations with very low values.

Authors reply:

Thanks for your suggestion, please note that this could occur as a result of several factors such as soil permeability, particle size, porosity, water content near saturation level, soil comprising of clay, exhalation, advection, temperature and diffusion.

Alonso, H., Rubiano, J. G., Guerra, J. G., Arnedo, M. A., Tejera, A., & Martel, P. (2019). Assessment of radon risk areas in the Eastern Canary Islands using soil radon gas concentration and gas permeability of soils. . Science of the Total Environment, 664(2019), 449-460. doi:. https://doi.org/10.1016/j.scitotenv.2019.01.411

2. The interpolation method (empirical bayesian kriging) is not described at all. Please give a brief description. Also, please elaborate why you have chosen this interpolation technique (which is rather novel and sophisticated) and not a simpler one such as a more basic kriging alternatives. What are the advantages of EBK?

Authors reply:

Thanks for your observation, please note that the description of EBK model, the explanation of why I chose this interpolation technique, has been added as a section. The advantage of EBK include: (a)It requires minimal interactive modelling (b) It gives more accurate standard errors of prediction than other kriging methods (c) It allows accurate predictions of moderately nonstationary data and (d) More accurate method for small data set than other Kriging methods.

Krivoruchko, K. (2012). Empirical Bayesian Kriging Implemented in ArcGIS Geostatistical Analyst. ArcUser Fall 2012 Software Development Team, Esri.

3. How was cross-validation conducted? Please give details on fold assignment. Did you consider spatial autocorrelation in the observational data during cross-validation? What is the range of spatial auto-correlation in your data?

Authors reply:

Thanks for your observation, please note that spatial autocorrelation was considered. The range of spatial auto-correlation in the data was inside the 95 percent interval.

4. You showed a dependence of GRP on geology. Why do you not exploit this knowledge for spatial prediction/interpolation, e.g. by using geology as co-variable?

Authors reply:

Thanks for your observation please note that, the EBK prediction/interpolation uses the geographical coordinates for the prediction/interpolation and it was done on a shape file of the geological map.

5. The conclusion is only partly supported by your results. The high risk of granitic areas seems to be rather your hypothesis than your finding. Large parts of the high GRP areas are not in the granitic areas. For instance, a large fraction of the area covered with quaternary sediments (in the south and south-west of the study area) has a higher GRP than the granitic areas. This might be a consequence of the interpolation technique that you selected by not implementing geology as co-variable.

Authors reply:

Thanks for your observation, please note that the conclusion has been rewritten to account for the outcropping of high GRP in quarternary and some low in granite areas.

6. In many cases the language is not precise enough, partly ambiguous and grammatically not always sound.

Authors reply:

Thanks for your observation, please note that the partly ambiguous and grammatical structures have been crossed checked and corrected.

Minor comments

• L50: Rn is one of the leading causes for lung cancer, not for cancer in general

Authors reply:

Thanks for your suggestion, please note that the sentence has been modified to reflect “one of the leading causes of lung cancer”

• L54: “generate higher indoor Rn levels”

Authors reply:

Thanks for your observation, please note that “generate higher indoor Rn levels” has been reframed to “the potential to generate higher indoor radon levels”

• L 56: these methods are not geology-based. I guess you mean statistical methods (geostatistics, regression) using co-variable/proxy data

Authors reply:

Thanks for your suggestion, please note that “geology based” has been corrected to “geostatistical-based”

• L 57: Ariel? Please explain

Authors reply:

Thanks for your observation, please note that “Ariel” was a typographical error and has been corrected to “Aerial” and has to do with the design of semiconductor and scintillation gamma ray sensors integrated into aerial robotic plat forms, for the surveillance in hard to reach, hazardous areas while allowing for the measurements to be dynamically tracked and mapped

• L76: What do you mean with “baseline” data? Unclear.

Authors reply:

Thanks for your observation, please note that “baseline” has been deleted so that the sentence will be clear.

• L 93: consider replacing geological “contexts” with “regions”

Authors reply:

Thanks for your suggestion, please note that “contexts” has been replaced with “regions”

• L96-98: The description of the sampling design is unclear to me. Please specify. Did you use the same fraction of measurements per geological region or did you sample preferentially in areas with expected higher values or higher geological heterogeneity?

Authors reply:

Thanks for your observation, please note that the sampling has carried out such a manner that each geological formation will be included. It was difficult to get uniform number of sampling per geological formation because of the challenge of accessibility of some locations and denial of permission by some property owners. 

• Fig.1: What is the scale of the map? Please add sampling locations. Your geological classification in the legend is a mix of stratigraphic and lithologic classes. Please be consistent. Either stratigraphic or lithologic or lithostratigraphic.

Authors reply:

Thanks for your suggestion, please note that the scale of the map has been inserted and sampling locations added to the map. The geological classification in the legend has been adjusted to lithologic.

• Please indicate sampling locations also in Fig. 3, 7 & 8. This will help interpreting the maps.

Authors reply:

Thanks for your suggestion, please note that the sampling locations have been indicated in Fig. 3, 7, & 8 as suggested.

• Fig 2: please consider using smaller bins (e.g. 5 or 10 kBq/m³) for the histogram. In the box statistical indicators, a median of 0.15 seems to be unrealistic given my experience and also the data shown in the plot. Consider using standard deviation instead of error since you want to give a measure of dispersion in your data set rather than an error/uncertainty of an individual observation. There is no boxplot in this figure (figure caption)

Authors reply:

Thanks for your suggestion, please note that smaller bins have been used for the histogram and standard deviation has been considered for measure of dispersion in the data. Box plot in figure caption has been deleted

• I would recommend not using two decimal digits because it pretends an accuracy that is not existing in reality

Authors reply:

Thanks for your suggestion, please note that the decimal digits have been adjusted.

• L132: A soil gas Rn concentration of 0.01 kBq/m³ is a very low value. This is rather a Rn concentration observed in outdoor air than in soil (where Rn is continouosly produced). I would suspect such low values as erroneous measurements (e.g., due to contamination of your measured volume with outdoor air). If you consider these values to be realistic please elaborate on the processes in soil causing such low concentrations.

 

Authors reply:

Thanks for your suggestion, please note that this could occur as a result of several factors such as soil permeability, particle size, porosity, water content near saturation level, soil comprising of clay, exhalation, advection, temperature and diffusion.

Alonso, H., Rubiano, J. G., Guerra, J. G., Arnedo, M. A., Tejera, A., & Martel, P. (2019). Assessment of radon risk areas in the Eastern Canary Islands using soil radon gas concentration and gas permeability of soils. . Science of the Total Environment, 664(2019), 449-460. doi:. https://doi.org/10.1016/j.scitotenv.2019.01.411

• L136: You performed log-transformation of your data prior to geostatistical modelling. How did you backtransform your data to the original scale after interpolation.

Authors reply:

Thanks for your observation, please note that the backtransform is done using the geostatistical analyst tool in the ArcGIS software

• L 166: “marine soils” are misleading for soils with marine sediments as parent material. Please consider revising.

Authors reply:

Thanks for your suggestion, please note that the line has been revised.

• Fig. 3. In this map a few apparent artefacts (e.g., in the east next to the label of Kelantan) are existent. Further, I suggest using even value breaks instead of giving two decimal digits.

Authors reply:

Thanks for your suggestion, please note that the map has been adjusted and the two decimal digits values has been changed.

• L 174ff: I suggest shifting the explanation of a boxplot into the caption of Fig4 because it breaks somewhat the flow in your text and it can be assumed that most readers are familiar with the concept of a boxplot.

Authors reply:

Thanks for your suggestion, please note that the paragraph has been matched with the previous one and the general description of boxplot has been moved to the materials and method as a sub-section.

• L180: The highest mean Rn conc. was found in Granite but the highest median in sed./metamorph. Please be more precise.

Authors reply:

Thanks for your suggestion, please note that the confusing sentence has been revised by removing “mediume” & “highest value”.

• Fig 4: Does Fig. show Rn data on the original scale or log-transformed data. If it shows data on the original scale, most values are exceptionally low which would require a physical explanation. How many samples do you have in each class? Please give n per class.

Authors reply:

Thanks for your observation, please note that Rn data was on the original scale. The physical explanation is that the low values are because of low soil permeability, high humidity and other meteorological factors. The number of samples have been indicated.

• L 211ff: the GRP is practically dimensionless since the log of k is used in eq. 1

Authors reply:

Thanks for your observation, please note that the dimension has been corrected

• L227: Please describe your geostatistical model. What are the parameters of your variogram?

Authors reply:

Thanks for your suggestion, please note that the geostatistical model and the parameters of the variogram has been describe in the sub-section Empirical Bayesian Kriging (EBK).

• L227: What is the reasoning for using 9 colour codes? If you do a classification in low, medium and high, it would be straightforward to use only 3 codes

Authors reply:

Thanks for your suggestion, please note that the classification low, medium and high has been changed to 3 colour codes

• L240: Please describe your cross-validation procedure. How did you assign observation to the testing fold? Random, stratified, spatially blocked?

Authors reply:

Thanks for your suggestion, please note that the cross-validation procedure that was used has been described in the section as suggested.

• Fig. 8: resolution of the figure is poor.

Authors reply:

Thanks for your observation, please note that the resolution of the figure has been improved.

• L259ff: your conclusion is not fully supported by your results. You claim that high GRP areas are mainly found in the granitic areas. In fact, the high GRP regions in Fig 7 and the outline of the granitic rocks (Fig1) do overlap only partly. I feel that this granitic regions being at higher risk is rather your hypothesis than your finding. Further, large parts of the high GRP areas are located in the quaternary sediments. This might be a consequence of their proximity to high GRP observations in the nearby granitic areas. If this is the case (I can only speculate on this issue because information on sampling locations is missing), I suggest considering the implementation of your knowledge (dependence of GRP on geology) in your spatial prediction/interpolation model (e.g., by using geology as co-variable).

Authors reply:

Thanks for your suggestion, please note that the conclusion has been rewritten to account for the outcropping of high GRP in quarternary and some low in granite areas.

• L 267: Why do you refer to a reference level of 300 Bq/m³? Your paper deals with the GRP, while the reference level of 300 Bq/m³ (or 100 Bq/m³or 200 Bq/m³) applies to indoor environments. The indoor reference level cannot be applied to your GRP estimates.

Authors reply:

Thanks for your observation, please note that the line has been reconstructed in terms of the GRP only.

 

Reviewer #2: 

Introduction

Lines 53-54: Please, check the definition of RPA, and clarify its relationship with indoor radon. “Radon Prone Area” is a term that describes an area where the probability of enhanced radon concentration is much higher than average. The establishment of such areas, by national authorities, can be based on geological conditions and/or the results of measuring campaigns of indoor radon concentration. See ICRP 1993 (Protection against radon-222 at home and at work. ICRP Publication 65. Paragraph 3.3)

Authors reply:

Thanks for your observation, please note that the sentence with the term “Radon Prone Area” has been reframed to contain “the potential to generate higher indoor radon levels”

Line 57: “Ariel” ?

Authors reply:

Thanks for your observation, please note that “Ariel” was a typographical error and has been corrected to “Aerial” and has to do with the design of semiconductor and scintillation gamma ray sensors integrated into aerial robotic plat forms, for the surveillance in hard to reach, hazardous areas while allowing for the measurements to be dynamically tracked and mapped

Lines 67-68: High GRP indicates a high availability of geogenic radon that can enters indoor. Also, express clearly that GRP is related to Ra content (but also U content!) because it is a Rn source, whereas permeability (but also faults!) is related to Rn mobility.

Authors reply:

Thanks for your suggestion, please note that the sentence has been reframed to include, “238U content” and “faults which are related to 222Rn mobility”.

Line 69: I suggest to add here a short review of the methods previously used to map GRP.

Authors reply:

Thanks for your suggestion, please note that a short review of the methods previously used to map GRP has been added.

Lines 75-76: Please, summarize the results of previous studies.

Authors reply:

Thanks for your suggestion, please note that summary of the results of previous studies are per the references numbers 26 – 33.

Line 78: What data? Please, add description and reference.

Authors reply:

Thanks for your observation, please note that it is referring to available data from literature on radon concentration studies in soil gas in Malaysia as per reference number 34.

Line 82: A strict definition of “radon risk” should be presented.

Authors reply:

Thanks for your suggestion, please note that “Radon risk” is herein defined as the probability or likelihood of harm or the severity of the harm from exposure to radon. The harm is mainly lung cancer, which is a painful and fatal disease

WHO. WHO Handbook on Indoor Radon: A Public health perspective. Geneva, Switzerland: WHO Press; 2009.

2.0 Materials and Method

In this section, after the description of the study area and the analytical techniques, I expected to read a paragraph about the geostatistical technique used to map the GRP. Practically, the discussion is based on the maps elaborated using Empirical Bayesian Kriging (EBK), but the method description is absent. Please, add the description of EBK model and explain why you chose this interpolation method. What semivariogram did you select? What the maximum number of points in each local model? What software did you use? What is the resolution of your maps?

Authors reply:

Thanks for your observation, please note that the description of EBK model, the explanation of why I chose this interpolation method, the semivariogram I selected, the maximum number of points in each local model, the software I used and the resolution of my maps has been added as a sub-section “Empirical Bayesian Kriging”.

2.1 Study area and sampling points

Lines 92-95: The geological description is too scarce, please add more information. Faults and structural discontinuities may be important pathways for radon movement in the subsoil, so you should describe the tectonic frame of the study area.

Authors reply:

Thanks for your suggestion, please note that more information has been added to the geological description. The tectonic frame of the study area has also been described as suggested.

Line 96-97: please, add the location of sampling points in Fig.1. Also, calculate the sampling density. How many samples for each geological formation?

Authors reply:

Thanks for your suggestion, please note that the sampling points have been added to figure 1. The samples in each geological formation were Quaternary (22), Sedimentary/Metamorphic rocks (26), Silurian (10), and Granite rocks (12) respectively.

2.2 Measurement of radon in soil gas

Please, specify in what period of the year and meteorological conditions the measurements were performed.

Authors reply:

Thanks for your observation, please note that measurements were carried out at low or no rainfall periods and moderate temperatures.

Line 109: Did you use a drying tube to maintain the relative humidity at low values?

Authors reply:

Thanks for your question, please note that the drying unit was used to maintain the relative humidity of the RAD7 at low values.

Results and discussion

3.1 Activity concentration of radon in soil gas

Lines 131-132: for non-normal distribution, better to use geometric mean

Authors reply:

Thanks for your suggestion, please note that the distribution standard deviation has been used instead in order to account for measure of dispersion.

Lines 132-134: this description is does not fit to the histogram of fig. 2, probably a histogram with more classes will be better describe your data. Please, consider using a normal probability plot; it can help to recognize both anomalous samples and multiple populations in log-normally distributed data

Authors reply:

Thanks for your suggestion, please note that a histogram with more bins have been used.

Lines 152-153: here the Authors introduce for the first time the Empirical Bayesian Kriging; see the comment above.

Authors reply:

Thanks for your observation, please note that Empirical Bayesian Kriging has now been introduced as a sub-section in the manuscript as earlier suggested.

Lines 153-155: this description is unnecessary; better to specify how you have determined the class breaks (e.g., manual interval, quantile, natural breaks etc.)

Authors reply:

Thanks for your observation, please note that lines 153 – 155 have been re-written in terms of how the class breaks were determined.

Line 157: “ Manjung District”, and other localities cited later in the text (e.g., line 214), are unknown to non-Malaysian. Please, add it in the Figures.

Authors reply:

Thanks for your suggestion, please note that “Manjung District” has replaced with the most western granitic region of the state”. It is not necessary to point out specific place. The rest of cited localities is removes and replaced with geological/cardinal direction.

Lines 158-159: An explanation is needed of how this list of soil types is used and how these soils were identified. Did you identified these soils in field? Did you extrapolated from a pedological map? Is there a correlation between soil types and Rn concentration and/or permeability?

Authors reply:

Thanks for your suggestion, please note that an explanation of how the list of soil types used and how these soils were identified has been added in the materials and method section.

Lines 159-162: This statement causes some doubts. The Authors identified the area with the highest values of soil gas radon in Manjung District, which is located in the southwestern part of the state of Perak; here, granite rocks outcrops and therefore the Authors explain the high Rn values because it derived from such bedrock. However, the geological map of fig. 2 shows that granite rocks extensively outcrop in the eastern part of the study area, where the soil gas Rn concentrations are not particularly high (Fig.3). So, how the Authors explain these spatial distribution? Did you consider the presence of permeable structures such as fractures and/or faults?

Authors reply:

Thanks for your observation, please note that this could occur as a result of several factors such as soil permeability, particle size, porosity, water content near saturation level, soil comprising of clay, exhalation, advection, temperature and diffusion. Moreover, most of the eastern part extensively outcropped by granite rocks were not accessible as the area is covered by thick forest and so were not sampled as seen in fig. 2 now that the sampling points have been indicated.

Line 163: What mines/quarries are in the study area? Where they are located?

Authors reply:

Thanks for your suggestion, please note that these were not considered in the scope of the study.

Line 169: I suggest to add a table (or a box-plot) with the statistics (average values, ranges of values) of soil gas Rn concentration and permeability in different soil types, I think that it can be help the reader.

Authors reply:

Thanks for your suggestion, please note that it is impossible to perform boxplot as the data per soil types (~ less than 7 samples per soil types) is too low, compared to geological influence which only 4 types. 

3.2 Radon activity concentration for each geological type

Sincerely, there is no reason to separate this paragraph from the previous one.

Lines 174-178: the general description of box-plot should be moved in the method section. A sub-paragraph describing statistical and geostatistical elaborations might be appropriate.

Authors reply:

Thanks for your suggestion, please note that the paragraph has been matched with the previous one and the general description of boxplot has been moved to the materials and method section.

3.3 Soil gas permeability

Lines 197-198: Please, add a map of spatial distribution of permeability, it could be very useful here. Please, also discuss how permeability influences radon in soil, in general and in your case study.

Table 1: Add the number of your samples that fall within each class of permeability.

Authors reply:

Thanks for your suggestion, please note that a map of spatial distribution of permeability has been added, and how permeability influences radon in soil has been discussed, and the number of samples that fall within each class of permeability has been added to Table 1.

3.4 Geogenic Radon Potential (GRP) and Radon prone areas

Line 210: “Many years of extensive research in the Czech Republic”…please, add references.

Authors reply:

Thanks for your suggestion, please note that references have been added.

Line 232: the title of the paragraph is “Geogenic Radon Potential (GRP) and Radon prone areas”, but where in the text the RPA are described?

Authors reply:

Thanks for your observation, please note that, “Radon prone areas” has been described already in L53 – 54 in the introduction.

Please, compare the map of GRP with the map of Terrestrial Gamma Dose Rate (TGDR) published by Ramli et al (2016) - Predicting terrestrial gamma dose rate based on geological and soil information: case study of Perak state, Malaysia. J. Radiol. Prot. 36(1), 20-36.

Authors reply:

Thanks for your suggestion, please note that it is impossible to compare two different measurement units between map. TGDR map produced by Ramli et al 2016 is an in-situ data of absorbed dose in air (nGy/h) meanwhile GRP map is a statistical score map. On top of that, 70 preliminary sampling data of the present study is incompatible to the comprehensive TGRD data survey by Ramli et al.

4.0 Conclusion

Lines 268-270: this concept should be emphasized in Discussion.

Authors reply:

Thanks for your suggestion, please note that concept in Lines 268-270 has been emphasized in the Discussion section.

Figures

Figure 1: add the location of sampling points, the prevailing tectonic lines, names of the district cited in the text

Authors reply:

Thanks for your suggestion, please note that the sampling points have been added

Figure 2: delete “and box plot” in the figure caption

Authors reply:

Thanks for your suggestion, please note that “and box plot” has been deleted

Figures 3 and 7: add the corresponding semivariograms.

Authors reply:

Thanks for your suggestion, please note that corresponding semivariogram has been added

Figure 4: what represent the blue dashed lines? How many samples for each geological type?

Authors reply:

Thanks for your observation, please note that the blue lines are value range of 222Rn activity ranges 0 – 1, 1 – 10, 10 – 100 and 100 and above. 

Figure 5: add box plots classified according to geological formations and/or soil types.

Authors reply

Thanks for your suggestion, please note that a box blot of soil gas permeability according to geological types have been added as figure 6.

Figure 7: add a layer of urbanized areas, in order to evaluate if the areas of high RI are already urbanized or not yet; in the first case, the map could indicate the areas where it could be necessary monitoring the indoor radon concentration, in the second case the map could be a tool for future urbanistic plans.

Authors reply:

Thanks for your suggestion, please note that for urbanised areas the indoor radon measurements are recommended.

---

## [Decision Letter · Decision Letter 1]

3 Mar 2021

PONE-D-20-31278R1

Soil gas radon and soil permeability assessment: Mapping radon risk areas in Perak State, Malaysia.

PLOS ONE

Dear Dr. Nuhu,

Thank you for submitting your manuscript to PLOS ONE. After careful consideration, we feel that it has merit but does not fully meet PLOS ONE’s publication criteria as it currently stands. Therefore, we invite you to submit a revised version of the manuscript that addresses the points raised during the review process.

ACADEMIC EDITOR comments:

While the authors have addressed minor issues with the original submission, there are still problems with this paper. Mapping Rn is the main objective of the paper so the mapping method must be explained and justified but Reviewer #1 points to lingering issues with this. The problem with the geostatistical method may need the authors to enlist help from a specialist in this field.

We look forward to receiving your revised manuscript.

Kind regards,

Elisabeth Bui

Academic Editor

PLOS ONE

Reviewers' comments:

Reviewer's Responses to Questions

**Comments to the Author**

1. If the authors have adequately addressed your comments raised in a previous round of review and you feel that this manuscript is now acceptable for publication, you may indicate that here to bypass the “Comments to the Author” section, enter your conflict of interest statement in the “Confidential to Editor” section, and submit your "Accept" recommendation.

Reviewer #1: All comments have been addressed

Reviewer #2: (No Response)

2. Is the manuscript technically sound, and do the data support the conclusions?

Reviewer #1: No

Reviewer #2: Partly

3. Has the statistical analysis been performed appropriately and rigorously? 

Reviewer #1: No

Reviewer #2: Yes

4. Have the authors made all data underlying the findings in their manuscript fully available?

Reviewer #1: No

Reviewer #2: Yes

5. Is the manuscript presented in an intelligible fashion and written in standard English?

Reviewer #1: Yes

Reviewer #2: Yes

6. Review Comments to the Author

Reviewer #1: Thanks to the authors for addressing all comments. Many of the minor issues related to grammar, style and figures have been mostly solved. However, the two fundamental issues of 1) Rn soil value range and the 2) the EBK predictive map are still unsolved from my point of view.

1. Soil Radon value range: I am still not convinced on the accuracy such low values (depth) which seem for most samples 1-2 orders of magnitude too low. You give median values (Fig. 4) between 0.1 and 1.0 kBq/m³ for the four different geological units. I have great doubts on the correctness of these data and rather suspect issues with your sampling procedure. I consider values below 10 kBq/m³ for soils with granitic parent material as rare and below 1 kBq/m³ as highly unlikely. In your study most of the values are below 1 kBq/m³. Based on my experience and the values reported in the literature these soils with granitic parent material are often reported to lie in the range 10 – 1000 kBq/m³. You correctly mention the various factors influencing soil Rn concentration such as soil physical properties, soil moisture, permeability etc. But, I cannot imagine that the interplay of these factors result in such a large number of very low values. Further, in contrast to your statement that low values may be associated with low permeability, I would rather expect the reverse pattern: low permeability decreases Rn transport (diffusion, advection) and, consequently, decreases Rn exhalation. This means at the same time increase of Rn accumulation in soils which is associated with higher Rn concentration.

I suggest participation in soil gas intercomparison measurements for quality assurance and to be able to exclude issues with your sampling procedure.

2. Your reasoning for selection of your mapping method EBK does not convince me. It seems that the selection of EBK was motivated by the ease of its implementation in the sense of an automatic data processing and map production. While this may be convenient, it is dangerous because the method seems to be rather treated as a black box.

You correctly emphasize the high importance of geology for spatial variability of Rn in soil. Giving this knowledge and the availability of the required data (geological map of your study region), why do you not implement this knowledge in your mapping procedure? At the current state, your interpolation completely ignores geological boundaries – you simply interpolate across those borders. Such a procedure cannot be assumed to provide accurate estimates since measurements in one geological unit cannot be used to produce predictions in another geological unit without consideration of geology as co-variable. From a geostatistical point of view, the range of spatial- autocorrelation varies throughout your study area. This issue manifests in the fact that your top hotspot region mainly results from a single very high value. This single high value spreads from the granitic area over a very large region into other geological units and just stops at the nearest observations. This is not plausible at all to me. I highly recommend implementing geology as co-variable for mapping soil Rn. This is feasible with EBK, but also with simpler geostatistical approaches (such co-kriging based on ordinary kriging).

Further, artefacts are still visible in the map in the north-east where no observations were conducted (I suggest to clip these areas because predictions are not supported by observations. This issue can also be seen in Fig. 11 (dark red region in the north-east). Moreover, the resolution (i.e., grid cell size, e.g. 1km * 1km) of the map is still unclear.

Reviewer #2: Dear Editor

The authors tried to respond the questions arose from the first review. However, some questions are still unresolved, therefore I propose a minor revision for further improving the manuscript. In detail:

L 86-87: Please, add a short description of the results of the cited references (references numbers 26 – 33) in the text of your manuscript.

L 169: Please, I would like to know the resolution of the data, i.e., the dimension of each pixel/cell of your raster (e.g., 10 x 10 km or 30" x 30")

L 203-205: you did not describe how the class breaks were determined. So, what classification method did you use (manual interval, equal interval, quantile, natural breaks, geometrical interval, standard deviation)?

L 302 and Fig. 9: You wrote “A three-color code was used in categorizing the RI.” So I imagine that you have used the classification described between the lines 286-290 (i.e., low RI (GRP < 10), medium RI (10 <grp>

Fig.1 Legend of geology: the lithology is still confused with the age of geological formations. The text (L 107-113) has the same problem.

Fig 2. I suggest to increase the number of bins (not only increase the number of classes in the X axis)

Fig 9: add a layer of urbanized areas or the location of the main cities. I agree that indoor radon measurements are recommended for the urbanized areas, I simply would know if there are important cities in the regions characterized by medium to high RI.</grp>

7. PLOS authors have the option to publish the peer review history of their article (what does this mean?). If published, this will include your full peer review and any attached files.

Reviewer #1: No

Reviewer #2: No

---

## [Author Response · Author response to Decision Letter 1]

10 Apr 2021

PONE-D-20-31278R1

Soil gas radon and soil permeability assessment: Mapping radon risk areas in Perak State, Malaysia.

PLOS ONE

ACADEMIC EDITOR comments:

While the authors have addressed minor issues with the original submission, there are still problems with this paper. Mapping Rn is the main objective of the paper so the mapping method must be explained and justified but Reviewer #1 points to lingering issues with this. The problem with the geostatistical method may need the authors to enlist help from a specialist in this field.

 Authors reply:

Thanks for your observation/suggestion, please note that the geostatistical method of mapping Radon has been changed to Cokriging based on ordinary kriging. An appropriate semivariogram general equation of the model has been included. 

 Review Comments to the Author

 Reviewer #1: Thanks to the authors for addressing all comments. Many of the minor issues related to grammar, style and figures have been mostly solved. However, the two fundamental issues of 1) Rn soil value range and the 2) the EBK predictive map are still unsolved from my point of view.

1. Soil Radon value range: I am still not convinced on the accuracy such low values (depth) which seem for most samples 1-2 orders of magnitude too low. You give median values (Fig. 4) between 0.1 and 1.0 kBq/m³ for the four different geological units. I have great doubts on the correctness of these data and rather suspect issues with your sampling procedure. I consider values below 10 kBq/m³ for soils with granitic parent material as rare and below 1 kBq/m³ as highly unlikely. In your study most of the values are below 1 kBq/m³. Based on my experience and the values reported in the literature these soils with granitic parent material are often reported to lie in the range 10 – 1000 kBq/m³. You correctly mention the various factors influencing soil Rn concentration such as soil physical properties, soil moisture, permeability etc. But, I cannot imagine that the interplay of these factors result in such a large number of very low values. Further, in contrast to your statement that low values may be associated with low permeability, I would rather expect the reverse pattern: low permeability decreases Rn transport (diffusion, advection) and, consequently, decreases Rn exhalation. This means at the same time increase of Rn accumulation in soils which is associated with higher Rn concentration.

I suggest participation in soil gas intercomparison measurements for quality assurance and to be able to exclude issues with your sampling procedure.

 Authors reply:

Thanks for your observation, please note that, the low Rn in soil values occurred as a result of an error in computing the average values from the raw data.

According to the RAD7 user manual, when radon is introduced into the chamber, at first there is no count. After a few minutes the counts begin in window A, which is polonium-218, a result of the decay of Rn-222. For the first five minutes or so the count rate increases, then begins to approach a steady level. After about 10 minutes, the polonium -218 daughter reaches close to equilibrium with the parent radon-222. The first two 5-minutes cycles should be ignored, while the next one or two cycles should be averaged to arrive at the radon concentration of the soil gas (Durridge RAD7 User manual, 2015): pp39 &67

In the earlier manuscript, the average was taken over six readings instead of four readings, as the first two readings are supposed to be excluded according to the RAD7 user manual. Find below a sample of the raw data printout in which the first two readings are 69.8 and 140 Bq m-3. respectively. An average of the six readings is 8710 Bq m-3 while excluding the first two readings and taking average of the four gives 13,013 Bq m-3 resulting to an underestimation of over 4,000Bq m-3. Thus, all the results have been recalculated accordingly.

2. Your reasoning for selection of your mapping method EBK does not convince me. It seems that the selection of EBK was motivated by the ease of its implementation in the sense of an automatic data processing and map production. While this may be convenient, it is dangerous because the method seems to be rather treated as a black box.

You correctly emphasize the high importance of geology for spatial variability of Rn in soil. Giving this knowledge and the availability of the required data (geological map of your study region), why do you not implement this knowledge in your mapping procedure? At the current state, your interpolation completely ignores geological boundaries – you simply interpolate across those borders. Such a procedure cannot be assumed to provide accurate estimates since measurements in one geological unit cannot be used to produce predictions in another geological unit without consideration of geology as co-variable. From a geostatistical point of view, the range of spatial- autocorrelation varies throughout your study area. This issue manifests in the fact that your top hotspot region mainly results from a single very high value. This single high value spreads from the granitic area over a very large region into other geological units and just stops at the nearest observations. This is not plausible at all to me. I highly recommend implementing geology as co-variable for mapping soil Rn. This is feasible with EBK, but also with simpler geostatistical approaches (such co-kriging based on ordinary kriging).

Further, artefacts are still visible in the map in the north-east where no observations were conducted (I suggest to clip these areas because predictions are not supported by observations. This issue can also be seen in Fig. 11 (dark red region in the north-east). Moreover, the resolution (i.e., grid cell size, e.g. 1km * 1km) of the map is still unclear.

 Authors reply:

Thanks for your observations/suggestion, please note that the geostatistical method has been changed to Cokriging based on ordinary kriging, and that geology has been implemented as co-variable for the mapping soil gas radon. The grid cell size of the map is 5km ×5km

Reviewer #2: Dear Editor

The authors tried to respond the questions arose from the first review. However, some questions are still unresolved, therefore I propose a minor revision for further improving the manuscript. In detail:

L 86-87: Please, add a short description of the results of the cited references (references numbers 26 – 33) in the text of your manuscript.

 Authors reply:

Thanks for your suggestion, please note that a short description of the results of references 26 -33 have been added in the revised manuscript. 

L 169: Please, I would like to know the resolution of the data, i.e., the dimension of each pixel/cell of your raster (e.g., 10 x 10 km or 30" x 30")

 Authors reply:

Thanks for your suggestion, please note that we have added the dimensions of the pixel in the revised manuscript. 5km ×5km

L 203-205: you did not describe how the class breaks were determined. So, what classification method did you use (manual interval, equal interval, quantile, natural breaks, geometrical interval, standard deviation)?

 Authors reply:

Thanks for your observation, please note that the class breaks were determined using the geometrical intervals and has been added to the revised manuscript. 

L 302 and Fig. 9: You wrote “A three-color code was used in categorizing the RI.” So I imagine that you have used the classification described between the lines 286-290 (i.e., low RI (GRP < 10), medium RI (10 

 Authors reply:

Thanks for your observation, please note that Figure 9 has been re-labelled to reflect low RI (GRP < 10), medium RI (10 >GRP) and high RI (35<GRP)

Fig.1 Legend of geology: the lithology is still confused with the age of geological formations. The text (L 107-113) has the same problem.

 Authors reply:

Thanks for your observation, please note that, addition details of the geological formations have been added in lines 107-113. 

Fig 2. I suggest to increase the number of bins (not only increase the number of classes in the X axis)

 Authors reply:

Thanks for your suggestion, please note that we have increased the number of bins in the Figure as suggested.

Fig 9: add a layer of urbanized areas or the location of the main cities. I agree that indoor radon measurements are recommended for the urbanized areas, I simply would know if there are important cities in the regions characterized by medium to high RI.

 Authors reply:

Thanks for your suggestion, please note that the location of main cities have been added to the figure.

---

## [Decision Letter · Decision Letter 2]

21 Jun 2021

Soil gas radon and soil permeability assessment: Mapping radon risk areas in Perak State, Malaysia.

PONE-D-20-31278R2

Dear Dr. Nuhu,

We’re pleased to inform you that your manuscript has been judged scientifically suitable for publication and will be formally accepted for publication once it meets all outstanding technical requirements.

Kind regards,

Elisabeth Bui

Academic Editor

PLOS ONE

Additional Editor Comments (optional):

A third independent reviewer has found this revision acceptable for publication. He has made some important comments regarding the difference in environmental conditions in Europe and Malaysia as they could affect the Rn gas transport process that I think would be worth mentioning in the discussion in your final version.

Reviewers' comments:

Reviewer's Responses to Questions

**Comments to the Author**

1. If the authors have adequately addressed your comments raised in a previous round of review and you feel that this manuscript is now acceptable for publication, you may indicate that here to bypass the “Comments to the Author” section, enter your conflict of interest statement in the “Confidential to Editor” section, and submit your "Accept" recommendation.

Reviewer #3: (No Response)

2. Is the manuscript technically sound, and do the data support the conclusions?

Reviewer #3: Yes

3. Has the statistical analysis been performed appropriately and rigorously? 

Reviewer #3: Yes

4. Have the authors made all data underlying the findings in their manuscript fully available?

Reviewer #3: Yes

5. Is the manuscript presented in an intelligible fashion and written in standard English?

Reviewer #3: Yes

6. Review Comments to the Author

Reviewer #3: Review of the paper PONE-D-20-31278R2 Soil gas radon and soil permeability assessment: Mapping radon risk areas in Perak State, Malaysia.

This publication is devoted to the important issue of ensuring the radiation safety of the population during exposure to radon in dwellings. One of the widely used tools for assessing the radon hazard of territories is the assessment of the geogenic radon potential. This approach has worked well in a number of European countries. For them, a good correlation was observed between the geogenic radon potential and radon concentration on the first floors of buildings and in detached houses. In this regard, it can be noted that the data on the mapping of such a state as Malaysia, for which data on the geogenic radon potential were obtained for the first time, are of undoubted scientific interest.

The work under review has been performed at a good technical level. The processing of the obtained experimental material was carried out in accordance with the approaches accepted in this field. The comments made by previous reviewers are answered in detail.

At the same time, the reviewer had one question. As you know, the geogenic radon potential takes into account two factors - the concentration of radon in the soil air and the permeability of the soil. With the same concentration of soil radon, but different soil permeability, the values of the geogenic radon potential may differ significantly. However, different values of soil permeability affect the radon flux from its surface only when there is a pressure gradient between the soil air and the air inside the building. In this case, the dominant mechanism of radon entry into the building is advective transport.Typically, this pressure gradient occurs when there is a temperature gradient between the indoor air inside the building and the outdoor air. In this case, the indoor air temperature should be higher than outdoor. The situation when the temperature of the air inside the building is higher than the temperature of the outside air is typical for most countries in Europe and North America. Therefore, the advective influx of radon from the soil into the building is considered dominant for low-rise and detached buildings. For Malaysia, the situation is somewhat different. It will be characterized either by the absence of a temperature gradient between the building and the outside atmosphere, or (when using air conditioners) by the presence of an inverse gradient when the temperature inside is less than in the outside atmosphere. Under these conditions, advective radon transfer will not be observed. The dominant mechanism of radon entry in this situation will be the diffusion mechanism. The rate of entry of radon will depend on the concentration of radium in the soil, the diffusion coefficient of radon, and the thickness of the soil layer to the groundwater level. Measuring these three parameters over a large area is unrealistic. According to the reviewer, the integral characteristic describing the diffusion entry is the radon concentration in the soil air, measured at the same depth. In this regard, it would be interesting to compare maps of soil radon concentrations and geogenic radon potential to assess the potential radon hazard of territories, taking into account various mechanisms of radon entry into buildings. This comment is a wish that does not affect the overall positive assessment of the work. I suppose that this work can be accepted for publication.

7. PLOS authors have the option to publish the peer review history of their article (what does this mean?). If published, this will include your full peer review and any attached files.

Reviewer #3: No

---

## [Editor Report · Acceptance letter]

19 Jul 2021

PONE-D-20-31278R2 

Soil gas radon and soil permeability assessment: Mapping radon risk areas in Perak State, Malaysia. 

Dear Dr. Nuhu:

I'm pleased to inform you that your manuscript has been deemed suitable for publication in PLOS ONE. Congratulations! Your manuscript is now with our production department. 

Kind regards, 

on behalf of

Dr. Elisabeth Bui 

Academic Editor

PLOS ONE